# Upscaling of a local model into a larger scale model

Vandenbulcke Luc[1,2] and Barth Alexander[3]

[1]seamod.ro, Jailoo srl, Romania
[2]MAST, Université de Liège, Belgium
[3]GHER, Université de Liège, Belgium

**Correspondence:** Luc Vandenbulcke (luc@seamod.ro)

**Abstract.** Traditionnally, in order for lower-resolution, global- or basin-scale (regional) models to benefit from some of the improvements available in higher-resolution sub-regional or coastal models, two-way nesting has to be used. This implies that the parent and child models have to be run together and there is an online exchange of information between both models. This approach is often impossible in operational systems, where different model codes are run by different institutions, often in different countries. Therefore, in practice, these systems use one-way nesting with data transfer only from the parent model to the child models. In this article, it is examined whether it is possible to replace the missing feedback (coming from the child model) by data assimilation, avoiding the need to run the models simultaneously. Selected variables from the high-resolution simulation will be used as pseudo-observations, and assimilated in the low-resolution models. The method will be called "upscaling".

A realistic test-case is set up with a model covering the Mediterranean Sea, and a nested model covering its North-Western basin. Under the hypothesis that the nested model has better prediction skills than the parent model, the upscaling method is implemented. Two simulations of the parent model are then compared: the case of one-way nesting (or stand-alone model), and a simulation using the upscaling technique on the temperature and salinity variables. It is shown that the representation of some processes, such as the Rhône river plume, are strongly improved in the upscaled model compared to the stand-alone model.

## 1  Introduction

In the present-day operational oceanography landscape, services are provided at different scales by different expert centers. At the European Union level, the Copernicus Marine Environment Monitoring Service (CMEMS) provides reanalyses, analyses and forecasts at global and basin scales. The models for the different basins are run by different institutes and centers within the regional monitoring and forecasting centers. Various oceanographic centers then use the CMEMS products to provide initial and/or boundary conditions to their respective models. These sub-regional and coastal models benefit from the specific knowledge of the local teams in their particular area of interest. Furthermore, nested models usually run at higher resolution, and may include more accurate data (bathymetry, river discharge data...) and processes of smaller scales, that cannot be easily included into basin-scale models. High resolution observations such as satellite sea surface temperature (SST), and recent ultra-high resolution products (see e.g. Le Traon et al., 2015) have been shown to be best assimilated into nested models, as chances are

higher that the observed processes are well represented (Vandenbulcke et al, 2006). Similarly, high-resolution observations of currents by high-frequency radars are expected to benefit most to models with a similar high resolution (i.e. nested models).

When parent and child models are run together (meaning, concurrently and on the same computing platform), it is possible to use two-way nesting; the benefits mentioned above of using a nested model are then transferred back to the basin-scale model. This has been shown numerous times in the literature, e.g. Barth et al. (2005); Debreu et al. (2012). The beneficial impact of the feedback from nested to the parent model is visible even outside the domain of the nested model. This constitutes the baseline hypothesis of the present study: it is desirable to "copy" the results of the nested model into the parent model.

To emulate this nesting feedback, missing in the operational context, it is analyzed whether results from the sub-regional model can be used as pseudo-observations and assimilated in the basin-scale model. Indeed, data assimilation is not limited to the use of (real) observations by measurement devices. Onken et al. (2005) used data assimilation as a substitute for one-way nesting in a cascade of nested models. Alvarez et al. (2000) used a statistical model to predict SST, which was then assimilated as pseudo-observations in a hydrodynamic model (Barth et al., 2006). In the proposed "upscaling" method, the pseudo-observations come from the nested model. From the point of view of the forecasting centers, a data assimilation scheme is already implemented in the basin-scale model. Hence, implementing the upscaling method requires only to obtain the high-resolution data and assimilate (parts of) it, along with the real observations, during the analysis phase of the system.

When using grid nesting, problems at the open boundary of the child model include stratification mismatches, artificial waves, artificial rim currents; and ultimately instabilities and model blow-up (Mason et al., 2010; Debreu et al., 2012). By upscaling the child model into the parent model, the latter will progressively gain consistency with the child model solution within its domain, being beneficial for the child model over time. Upscaling can potentially reduce the risk of discrepancies at the open sea boundary.

Upscaling can also be seen as using a high-resolution model as a "measurement device" that replaces ever-too-sparse (real) measurements. Guinehut et al. (2002, 2004) showed that a coverage of the North Atlantic with a $3°$-resolution grid of Argo floats allows to effectively represent the large scales. Using a $5°$ array reduces the precision of the estimated fields two times. Currently, some CMEMS areas are largely undersampled.

Upscaling can be understood as a complement to downscaling (initialization) techniques such as presented in Auclair et al. (2000, 2001) (VIFOP) or in Simoncelli et al. (2011). The point of these methods is to combine interpolated fields coming from the large-scale model (the background or first-guess field) and existing high-resolution fields, so that small-scale structures present in coastal models are not lost whenever it is (re)-initialized by fields interpolated from the basin-scale model, and the obtained fields are physically balanced with respect to the coastal physics. If upscaling is used to improve the basin-scale fields and accord them with the coastal model, the "first guess" will be already much better.

Schulz-Stellenfleth and Stanev (2016) is another recent example showing the benefits of two-way nesting, especially in sophisticated modern-day forecasting systems. The study demonstrates that two-way nesting is critical for correct energy transfers between large and small scales (especially in coupled ocean-wave-atmosphere models) for cross-border advection, for the correct use of high-resolution coastal observations that cannot be fed directly into a large-scale model, etc. Ackowledging that operational systems are using only one-way nesting, Schulz-Stellenfleth and Stanev (2016) therefore strongly advocate the research into "upscaling" techniques. The present article tries to develop precisely such a technique.

In this article, the upscaling procedure is tried out in a realistic, nested model configuration covering the Mediterranean Sea and the North-Western basin and simulating the year 2014. The same model, NEMO 3.6 (Madec, 2008), and the same vertical resolution, are used for both configurations (only the horizontal grid is different). It is not expected that the conclusions of the study would be fundamentally different if different models and vertical grids are used for parent and child models. If different model codes were used, they could represent different processes. Hence, this should be taken into account by modifying the (representativity part of the) observation error covariance matrix when performing the data assimilation of the pseudo-observations. Examples of such contributions to the representativity error could be

- different vertical coordinates (see above)

- different implementations of the ocean surface: rigid lid or free surface; for the latter, linear or non-linear representation

- hydrostatic model or not

- different atmospheric forcing fields

- different turbulent closure schemes

- different numerical schemes for advection, horizontal diffusion etc.

The most striking difference between the parent and child models however, remains the horizontal resolution, and therefore, the general conclusions of the paper are expected to be valid, and upscaling should not be limited to the case of parent and child models being identical.

In this study, it is not the aim to verify that the nested model is indeed more realistic, according to some metrics, than the parent model. Rather, this consistutes the baseline hypothesis, and thus it is always considered beneficial to bring the parent model simulation closer to the child model simulation. It should be noted that some high-resolution processes, resolved by the nested model but not by the parent model, could have large phase errors in the nested model. In this case, the baseline hypothesis would be violated, and the nested model could actually have higher errors than the former. This aspect is not considered in the paper.

The parent model (also called "low-resolution" model), and the child (or "nested" or "high-resolution") model, and the data assimilation scheme are described in section 2. The parentmodel will be run both in "free" mode, and "upscaled" mode (i.e. assimilating pseudo-observations from the child model). The "free model" is equivalent to a stand-alone model, i.e. even if there are nested models, it is not influenced by them. Section 3 proposes some metrics to evaluate the system, related to the Rhône river plume, the cross-shelf exchanges, the large-scale current, SST, and the formation of Western Mediterranean Deep Water. Results are given in section 4 and a conclusion in presented in section 5.

## 2   Model and data assimilation configuration

### 2.1   Hydrodynamic model

The upscaling technique has been implemented in the North-Western Mediterranean Sea (NW-Med), including the Gulf of Lions and the Ligurian Sea (see Fig. 2). The region is characterized by large-scale currents (the Northern Current also called Liguro-Provencal Current, created by the junction of the Eastern and Western Corsican Currents, see e.g. Pinardi et al. (2015)), by intense meso-scale activity and by inertial oscillations. Furthermore, the NW-Med is the siege of formation of Western Mediterranean Deep Water (WMDW), important to the circulation in the whole Mediterranean Sea (e.g. Millot, 1999; Pinardi et al., 2015; Bosse et al., 2015; Somot et al., 2018; Simoncelli et al., 2018).

A realistic, one-way nested configuration was implemented using the NEMO 3.6 model and the AGRIF nesting tool (Debreu et al., 2008), covering respectively the Mediterranean Sea (MED) with a 8 km horizontal resolution, and the North-Western Mediterranean basin (NW-Med). The parent model resolution is similar to the previous version of the CMEMS Mediterranean Sea analysis-forecasting system (up to October 2017) (Clementi et al., 2017) and to the present reanalysis (1/16°) (Simoncelli et al., 2014). The child model horizontal resolution is 1.6 km.

When implementing the upscaling method, it is expected that after some time, the feedback from the NW-Med model will modify the Northern Current position and intensity in the parent model, which will in turn influence the NW-Med model through its open-sea boundary. The boundary condition provided by the MED model also influences the stratification of the water column, which is important for the pre-conditionning of the convection (S. Somot, private communication).

Both model bathymetries are interpolated from the GEBCO bathymetry. Thanks to its higher resolution, the bathymetry of the nested model (1/80°) is more realistic than in the parent model, at the coastline and more importantly, at the different canyons at the Gulf of Lions shelf break. The temperature and salinity initial condition is interpolated from the CMEMS Mediterranean reanalysis (1/16°) for 01/01/2014 (see https://doi.org/10.25423/medsea_reanalysis_phys_006_004), using tri-linear interpolation and linear extrapolation where needed. The model starts from rest. Atmospheric fluxes are computed using the bulk formula from the Nemo MFS module; the atmospheric forcing fields are obtained from ECMWF ERA Interim with a temporal resolution of 3 hours and a horizontal resolution of 0.75° reinterpolated by the ECMWF server to 0.125° (Dee et al., 2011). In the MED model, the flow between the Black Sea and the Mediterranean Sea, through the Marmara Sea and

the Dardanelles Strait, is modelized as a river, using climatological flow, temperature and salinity values. The salinity of the incoming water has a minimum and maximum of 22.5 psu and 27.5 psu reached in July and March respectively. Five other rivers (Rhône, Po, Ebro, Nile, Drin) are also represented, and monthly climatologic values for the flow and temperature are used, whereas the salinity is put to 5 psu, except for the Drin river where is it put at 2 psu. Using climatologic monthly values is coherent with the operational set-up in the CMEMS Mediterranean system, although the latter represents many more small rivers.

Daily Rhone river discharge measurements at the Beaucaire station were obtained from the Compagnie du Rhône, in order to be used in the nested NW-Med model. Interestingly, the total annual flow computed from the climatology and from the measured values for 2014-2015 are very similar (1% difference). However seasonal and daily values can be very different (see Fig. 1a). In particular, during the considered period, the climatology underestimates the winter discharge, but overestimates the summer discharge. Hence, depending on the dataset used, it is expected that the modelled river plume will also be significantly different. This is illustrated in Fig 1b, showing the surface salinity difference using climatological or daily discharge data in the child model (NW-Med) after 1 month of spin-up. The plume obtained using real river discharge extends much further offshore, almost completely across the Gulf of Lions, whereas the plume obtained with the climatological river discharge is essentially staying at the coast close to the river mouth. This is consistent with the much larger (almost double) discharge values observed in the real river data during January 2014.

## 2.2 Upscaling experiment description, Ensemble generation, and Data Assimilation scheme

In order to assimile pseudo-observations into the basin-scale models, different setups could be implemented, regarding the choice of the pseudo-observations, the frequency of assimilation, the data assimilation scheme itself, etc. The choices described below are consistent with current-day practices in the CMEMS operational systems. In particular, none of them currently assimilates velocity fields, and all of them use parameterized model state vector error covariances. Only one system (the Arctic system) currently uses an Ensemble Kalman filter, but the other systems are planning to evolve toward ensemble simulations in the future.

The following settings were chosen for the current experiment. The filter will be an Ensemble Kalman filter (the parent model is thus transformed into an ensemble of models). Assimilation will be performed daily. Only temperature and salinity will be used as pseudo-observations; the thinned 3D fields will be used. Velocity and surface elevation are not updated by the data assimilation procedure. Thinning is realized by taking the average of 5x5 cells of the nested model. The thinned pseudo-observations coming from the nested model are then considered independent, i.e. their error covariance matrix is diagonal. This is still a strong assumption which should be taken into account when determining the (diagonal) part of the observation error covariance matrix.

The members of the ensemble have perturbed initial conditions, atmospheric forcing fields and Rhône river discharge, similar to Auclair et al. (2003). The initial condition is the randomly weighted sum of the real initial condition (01/01/2014), and 6 other initial conditions (1 year, 20 days and 10 days earlier, and 10 days, 20 days and 1 year later). The weight of the real

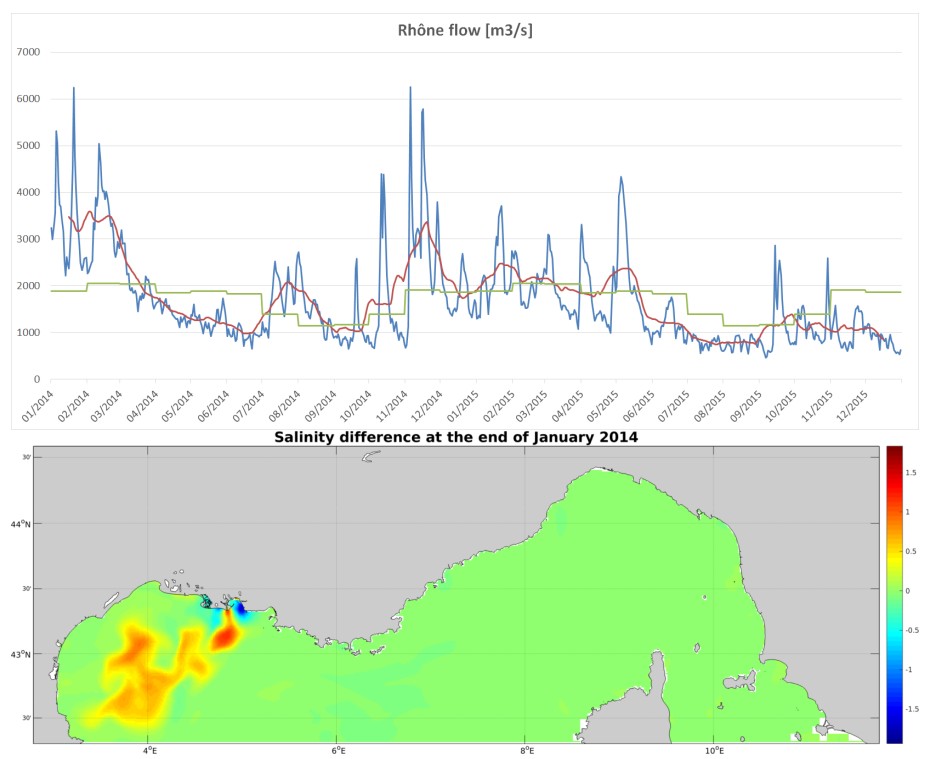

**Figure 1.** (above) Rhône river discharge from (green) the climatology, (blue) the measurements by the Compagnie du Rhone at the Beaucaire station, (red) the 1-month moving average of the measurements. (below) Difference of surface salinity in 2 different model runs of the nested grid on 28/Jan/2014, when using climatological or measured Rhône discharge values (i.e. using the green or blue curve in the upper panel)

.

initial condition is a random-normal number chosen in the Gaussian distribution with mean 0.5 and standard deviation 0.2; if necessary, the random number is then limited back into [0.2 0.8], whereas the 6 remaining weights are random numbers chosen uniformly in [0 1], and normalized so that the sum of all 7 weights is 1. This procedure ensures that the stability of each member is not modified (for example, the linear combination of 7 stable water columns is still a stable water column).

5   The atmospheric forcing fields of air temperature at 2m height and wind speed at 10m height are perturbed following the same procedure as in Barth et al. (2011); Vandenbulcke et al. (2017). Point-wise, the forcing fields are decomposed in Fourier series (from 3 hours to 1 year). For each member, a random field is generated, using these Fourier modes and random coefficients which have a temporal correlation length corresponding to the respective mode. This random field is added to the original field. The Rhône river discharge is perturbed using a random walk approach, with the expected perturbation after one year set as

10   20%. The other rivers are outside the observed part of the domain, and their discharge is not perturbed. With all 3 perturbations, an ensemble of 100 members is then spun up for 1 month.

Data assimilation is performed by the Ocean Assimilation Kit (OAK) package (Barth et al., 2008) implementing different filters such as SEEK and the Ensemble Kalman filter (EnKF). Different variants of the EnKF exist, and are classified in stochastic and deterministic methods. The former require to perturb the observations, adding sampling noise. The latter, also called Ensemble Square Root Filters, do not present this requirement; the perturbation approach is only applied in the model to obtain model errors. Different variants are compared in Tippett et al. (2003). One variant, called the Ensemble Transform Kalman Filter (Bishop et al., 2001; Wang et al., 2004), is used in this study. The filter equations are listed in Barth and Vandenbulcke (2017).

Nerger et al. (2012) summarizes how the spurious long-range correlations can be suppressed using so-called covariance localization, or domain localization and its addition observation localization. OAK uses the latter variant introduced in Hunt et al. (2007). In essence, the state vector is split into subdomains (water columns). In every water column, the analysis is performed independently (domain localization). In addition, for every water column, only nearby observations are used and the inverse of their error variance is multiplied by a localization function (observation localization). In the current setup, the localization function is a radial Gaussian function with an e-folding distance of 30 km.

The observation errors for temperature and salinity are set respectively at $0.3°C$ and $0.09$ psu. These values were determined after a sensitivity experiment with observation errors of ($0.5°C$, $0.15$ psu), ($0.3°$, $0.09$ psu), ($0.2°C$, $0.05$ psu) or ($0.1°C$, $0.03$ psu); as a trade-off between generating a close emulation of two-way nesting (hence very small observation errors), and generating fields as balanced as possible, that will not cause adjustment shocks into the model (hence larger observation errors). With the latter 2 choices for the observation error, the obtained assimilation increment was not much larger than with the final choice of ($0.3°$, $0.09$ psu), but qualitatively, unrealistic small scale variations started to appear.

From a technical point of view, OAK allows to use a multi-variate multi-grid state vector. As the Mediterranean model is parallized in 64 tiles, the multi-grid feature allows to update directly the tiles from the Mediterranean model restart files, influenced by the nested model, without including the other tiles in the state vector. The procedure thus allows to skip the reconstruction of the complete Mediterranean restart files. It should be noted that the tiles of the parent model, considered in the data assimilation procedure, are the ones covering the nested-model area, but also the neighbouring ones which are influenced by the data assimilation.

## 3   Metrics

To assess the upscaling method, five metrics were defined, that allow to compare the nested model and the parent model, in both cases without upscaling (MED free model) and with upscaling (MED upscaled model). If upscaling is succesfull, the parent model with upscaling will be closer to its nested model, than its counterpart without upscaling.

### 3.1   Cross-shelf transport

The penetration of off-shore water on the GoL (or inversely when negative transport values are obtained), is critical for the circulation on the shelf, for the shelf-open sea exchanges, etc. It is obtained by integrating the current over the boundary shown

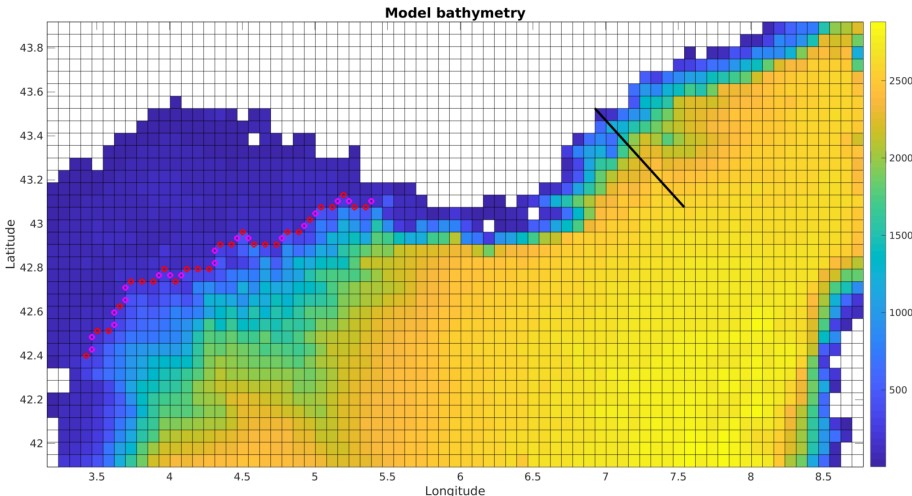

**Figure 2.** Zoom of the MED model grid, with the positions for computing the Northern Current metric (black line) and the cross-shelf transport metric (magenta and red dots)

in Fig. 2. This metric is useful to compare basin-scale models (free and upscaled) and check whether the upscaling procedure is able to drive the solution toward the nested model solution. The intensity of the cross-shelf transport cannot, however, be compared to real measurements (by lack of them).

## 3.2 Northern Current intensity

The Northern Current (NC) is the most important large-scale feature of the region of interest. It is considered to have a width of 40-50km during summer and 20-30km during winter; but the most offshore currents do not modify the transport much. Similarly, the NC is considered to be 100-200m deep in summer and 250-400m in winter. Following Alberola et al. (1995), its intensity is obtained by integrating the currents normal to a line from Nice to the location (43.0756°N, 7.5415°E), 214 km to the South-East, indicated in Fig. 2. As for the previous one, this metric only allows to inter-compare different models.

## 3.3 Rhône river plume

The plume of the Rhône River is measured by selecting all points around the river mouth with a salinity smaller than 37psu, and then choosing the most distant one from the river mouth. This provides the plume length and direction, although it may be an approximation: the plume can be curved, in which case its real length is larger than the estimation, or it can cover a large area, in which case the algorithm still obtains an azimuth although in reality it is not well defined.

This metric can be used quantitatively to compare models. Furthermore, it can be used to compare model results to real measurements. Indeed, although the real Rhône river plume length and direction are not measured directly, they can be estimated from satellite chlorophyll-a images. The model-observations comparison is then qualitative.

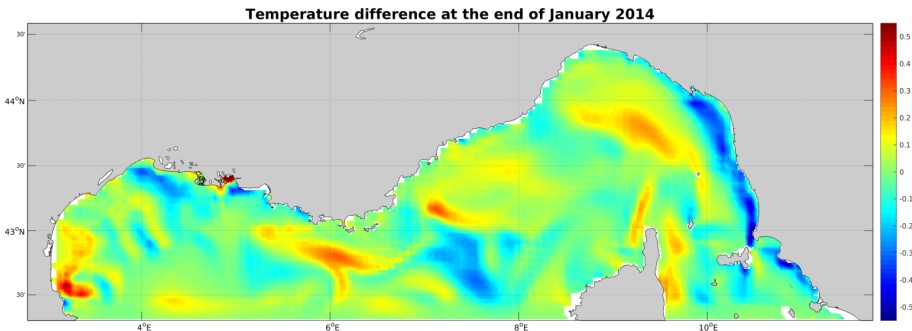

**Figure 3.** Temperature difference between the parent and nested models at 31/01/2014, projected on the nested model grid.

### 3.4 Sea surface temperature

This metric is the root mean square (RMS) difference between the parent model and the observed SST. For the latter, the L3 images are used.

### 3.5 Western Mediterranean Deep Water formation

Following Bosse et al. (2015) and references herein, the formation zone of Western Mediterranean Deep Water (WMDW) is comprised in 41-43°N, 4-6°E. WMDW forms an easily identifiable water mass: it has a temperature between 12.86~12.89°C, a salinity of 38.48~38.50 psu, and its depth is larger than 1000 m. The nested model (NW-Med) southern boundary is at 42.3°N, and hence only a part of the formation area is included in the area of MED covered by pseudo-observations. The WMDW metric measures the total volume [m$^3$] of WMDW in the domain covered by the NW-Med model, and is used to compare the different models.

### 4 Results

The temperature difference between the (unperturbed) parent and child models at the end of the spinup (31 January 2014) is represented in Fig. 3 on the child model grid. There are large temperature differences at the shelf break of the Gulf of Lions (the canyons are much better represented in the nested model); which extend all the way from the surface to the bottom of the Gulf of Lions. Other large differences appear in the Eastern and Western Corsican Currents, and their junction resulting in the Northern Current, as well as at the southern open boundary. The difference in salinity (not shown) has large values around the Rhône river plume (over 1 psu), and in a lesser extent in the Eastern Corsican Current. It appears that after a month, the differences are already significant, and if one trusts the nested model more, then it would be beneficial to bring these differences back to the basin-scale model.

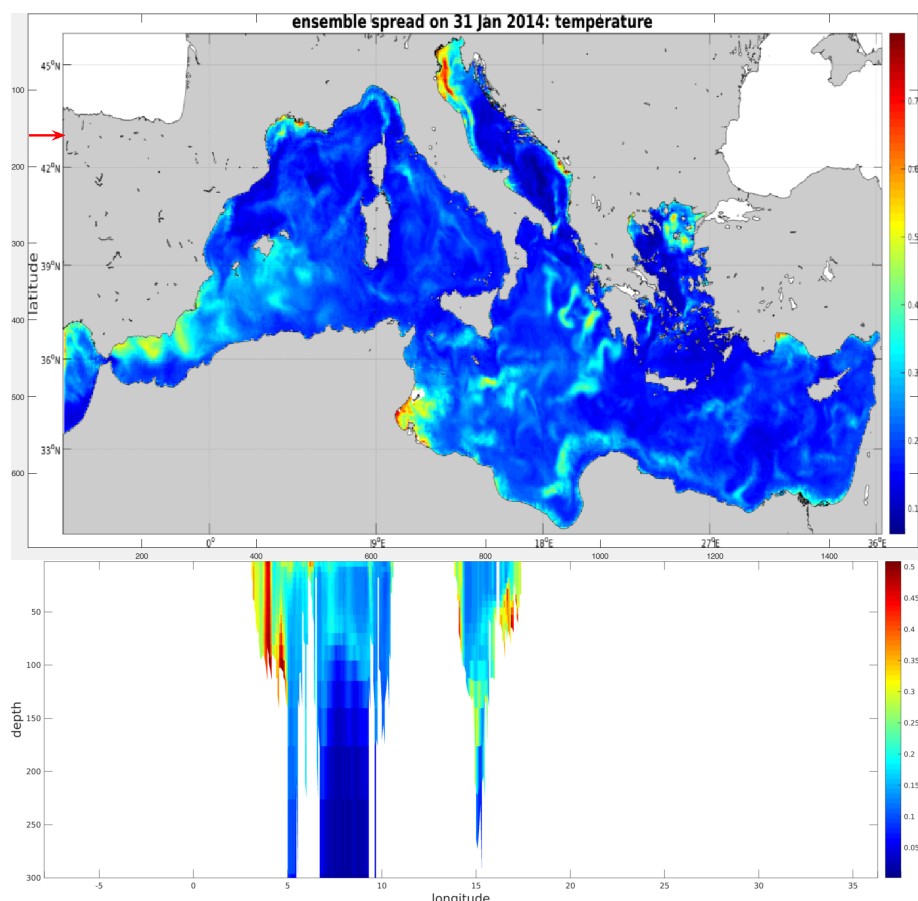

**Figure 4.** Spread of the ensemble of MED models at 31/01/2014: (upper panel) surface temperature, (lower panel) section at 43°N, indicated by a red arrow on the upper panel

At the end of the spin-up, the spread of the ensemble of models (Fig. 4) is very visible over the basin, at all river mouths, but also in other areas (Alboran Sea, Tunesian coastal zone...) as all 3 perturbations are applied at once. The ensemble spread is also visible in depth (i.e. deeper than when only the river discharge is modified).

As an example, the first data assimilation cycle is shown in Fig. 5 depicting SST. The L4 SST image is shown only for visual comparison. Qualitatively, it appears that upscaling changes important features: the Rhône river plume is oriented offshore instead of being mostly along-shore; fronts seem to be more well-defined; and the Northern Current flows along the shelf break instead of covering a large part of the shelf. The nested model, and the "upscaled" model, seem to be in closer agreement with the satellite image, than the free model.

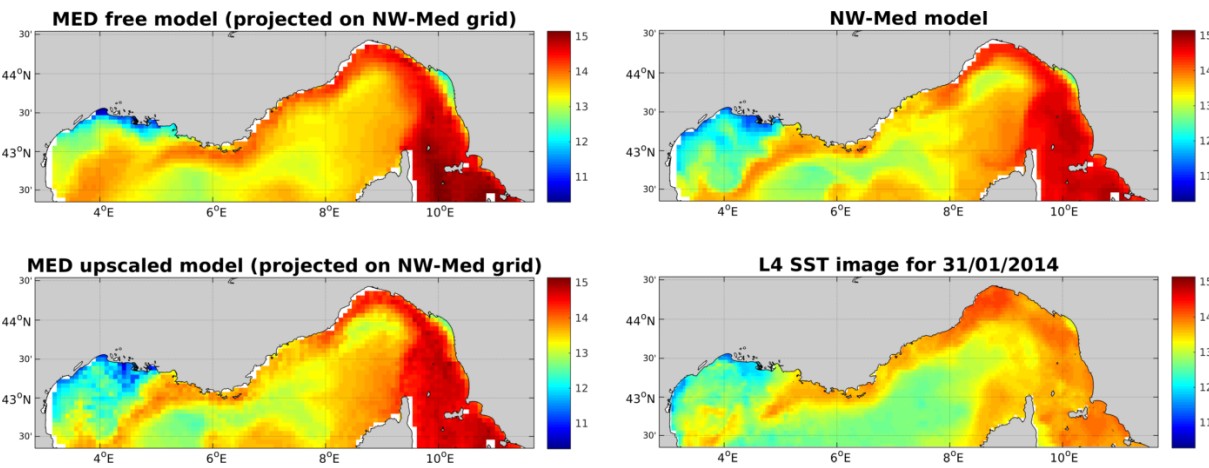

**Figure 5.** Sea surface temperature after the first upscaling step (31/01/2014), in the free model (upper left), nested model (upper right), upscaled model (lower left) and L4 satellite observation (lower right)

## 4.1 Cross-shelf transport

The flow accross the shelf break is represented in Fig. 6, for the basin-scale model in the free and upscaled cases. Although alternating periods of inflow and outflow appear, the transport seems to show a chaotic behavior. Yet it can be seen that while both models are generally similar, some periods exist where the simulated transport is very different. During the first month (February 2014), the free model predicts a net outflow during the first 2 weeks, followed by a net inflow during the last 2 weeks. The nested model (not shown) and hence also the upscaled model actually predicts the exact opposite. The reasons for the nested model to behave differently than the parent model may be an effect of wind interaction with the (different) bathymetries, or related to the different resolution. The actual transport is not measured or available; but the result of interest here is that the upscaling method is able to align the (parent model) currents with the ones from the nested model, and hence emulate two-way nesting, although only temperature and salinity pseudo-observations are used.

During the remainder of the year, the upscaled model predicts somewhat larger transports (both inward and outward). Generally speaking however, the two transport curves are closer than in February (or at least they are not of opposite signs anymore). Noticeably, in August-September, the upscaled model predicts a period of large inflow on the Gulf of Lions. The free model also predicts this inflow, but delayed by about 2 weeks.

The RMS difference between the parent and child models is shown in Table 1, for the MED free model and the MED upscaled model.

## 4.2 Northern Current

The transport by the Northern Current off Nice is represented in Fig. 7. Over the whole period, the root mean square difference between parent and child models is 0.22 Sv for the free model, and 0.19 Sv for the upscaled model. The same qualitative ob-

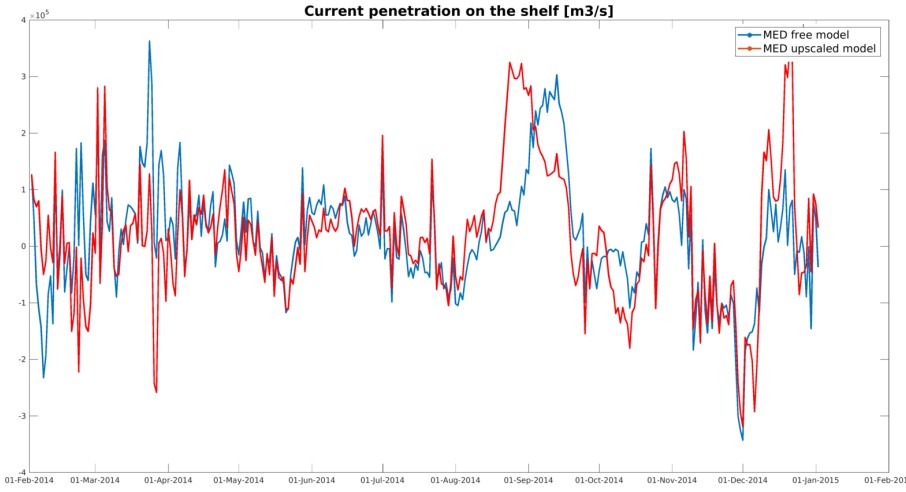

**Figure 6.** Water transport accross the shelf break during 2014, as obtained by the free (blue curve) and upscaled (red curve) parent models. Positive values indicate a net on-shelf transport.

servations can be made as for the cross-shelf transport. Both models generally agree, but periods exist with relatively important differences. Interestingly, a large difference appears in August-September, when the free model predicts a larger transport than the upscaled model. This is also the period when the transport accross the shelf break presents a temporal shift in between models.

5 For the purpose of our study, this metric cannot be used to validate the model since real measurements of the Northern Current transport are not available; but (as for the previous metric), it can be used to compare models, and to show that our goal is reached and upscaling of scalar fields is able to modify the velocity field of the parent model although only temperature and salinity are observed. The RMS differences between parent and child models are again given in Tab. 1.

### 4.3 River plume

10 The Rhône plume is perhaps the feature most significantly altered by upscaling. During the first month of the upscaled simulation, the free parent model usually places the plume along-shore, to the North-West, whereas the child model (and the upscaled parent model) usually orient the plume off-shore to the South-West (see Fig. 8). On top of the resolution-related differences between parent and nested models (in particular the bathymetry and the interaction of the water masses with the wind), both models have different freshwater discharge values, which is usually much higher and has also a much larger variability in the 15 nested model during February 2014. The upscaling method is clearly able to make the parent model ingest the different plume dynamic coming from the nested model. During another period (late August - early September), the opposite case occurs: the free model plume is oriented off-shore, but the nested (and upscaled) model predicts an along-shore plume. Apart from these 2 periods, differences between parent and child models are smaller; therefore, the time-average of RMS difference between the parent model and the nested model length is reduced only from 95.1 to 88.0 km (see Tab. 1).

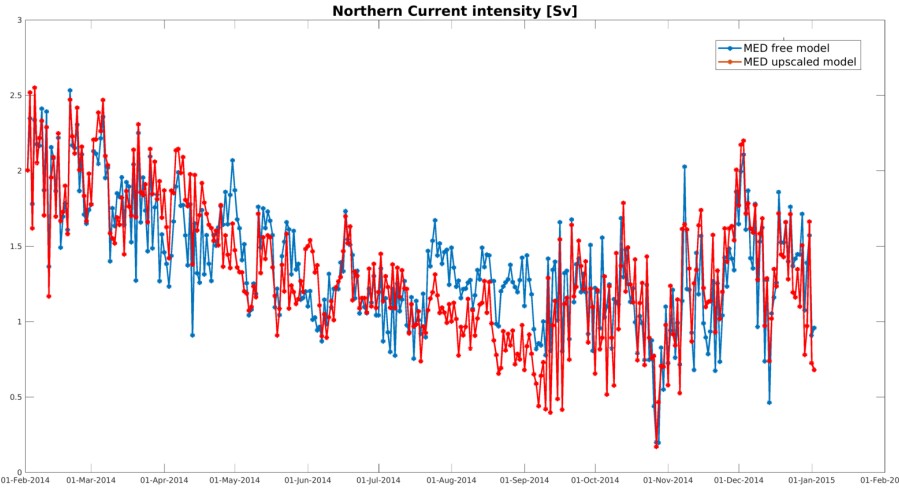

**Figure 7.** Water transport by the Northern current off Nice (France) during 2014, as obtained by the free (blue curve) and upscaled (red curve) parent models.

As a side note, the river plume can qualitatively be compared to real observations by using satellite observations of chlorophyll. During the first month of simulation, where the most significant differences appear, only a few level-3 satellite images are not almost entirely obscured by clouds. An example is given in Fig. 9 for 12 February 2014. One can clearly see the off-shore plume from the chlorophyll observations, whereas the free model plume is mostly along-shore. The nested (not shown) and
upscaled models correctly place the plume off-shore.

## 4.4 SST

The sea surface temperature metric allows to quantify the model error by comparison with satellite images. Level 3 images are used for computing the metric. The RMS difference between the different models and the L3 image are shown in Fig. 10
for the first 2 months of simulation. It appears that the RMS difference is around 0.4-0.5°C, even though no data assimilation is performed. Usually, the nested model is better still in some areas (e.g. coastal waters), and the upscaling procedure brings back these local improvements to the parent model (see Fig. 5 for an example). However, the area-wide RMS error is not influenced very much by upscaling (see Tab. 1), as large areas are essentially unmodified (parent and child models use the same atmospheric forcing fields and the same bulk formulae).
Some days, some processes appear to be missed by the models (both parent and nested), so that the RMS error is relatively large. In this case again, upscaling does not influence the RMS error of the parent model very much, as the nested model is not representing these processes any better than the parent model.

In both cases, this does not imply that the upscaling method is flawed, but rather that, in the current setup, the nested model is not able to generate an RMS error significantly lower than the parent model; hence upscaling does not have much to feed on.

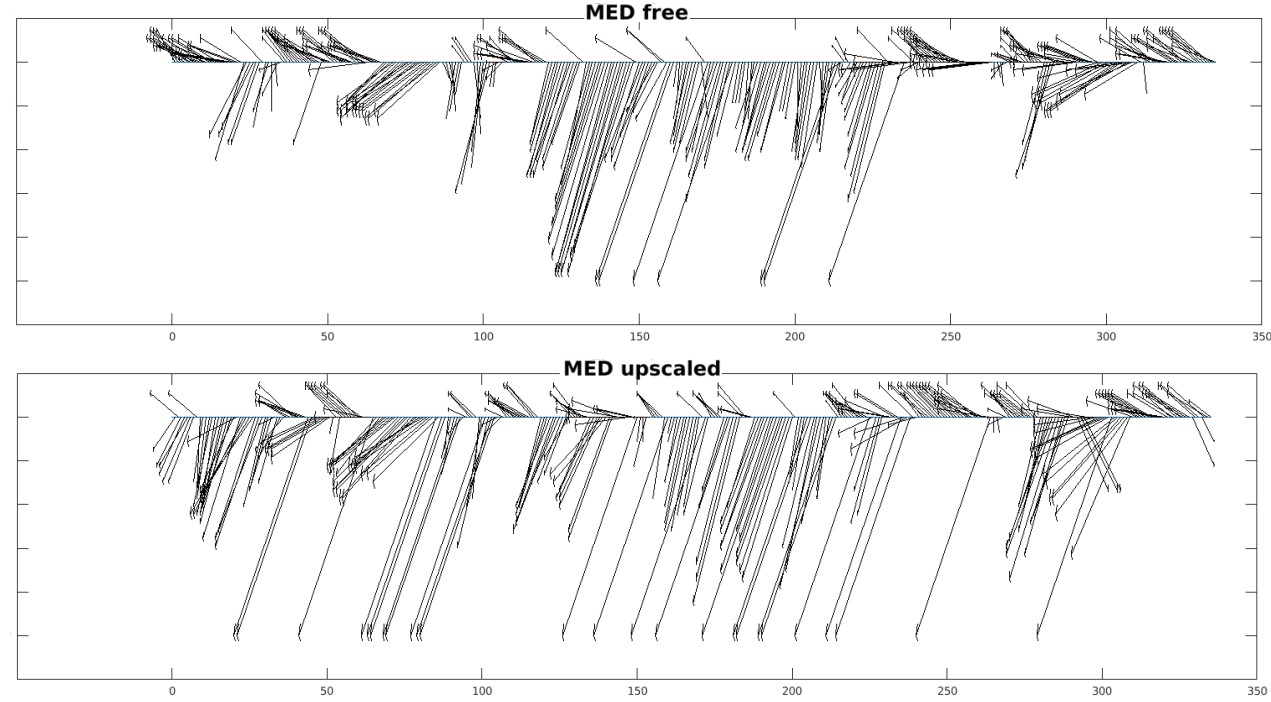

**Figure 8.** Rhône river plume direction and length (upper panel) for the free and (lower panel) upscaled MED models. The horizontal scale represents the days after the start of the upscaling experiment (Feb/2014).

Fig. 10 shows the RMS error during the first 2 months of simulation. The situation worsens during summer when the computed RMS errors are of 3°C) both for parent and child model; the difference in between models is hidden by the temporal variability of the error (not shown). In any case, the upscaled model is still very close to both the (free) parent and the nested models.

## 4.5 WMDW

5    The total amount of Western Mediterranean Deep Water in the free model (blue curve in Fig. 11) and the nested model (green curve) is periodically important ($10^3$km$^3$), but models do not converge during the simulation, as it appears during most of the second half of the year. Upscaling largely modifies the parent model, which in turns provides modified boundary conditions to the nested model, so that after a while, the upscaled model and its child model significantly diverge from the free models. Without measurements and due to the choice of the model domain, it is not possible to assert which pair of models is more
10   realistic. However, as for other metrics, the discrepency between parent and child model is reduced in the upscaled pair of models, which is certainly a desirable characteristic (see Tab. 1). This can be explained by the fact that the data assimilation also modifies the parent model solution outside the nested area (in the limit of the localization radius used in the data assimilation procedure). Therefore, the water immediately outside the nested domain is modified and made more coherent with the nested

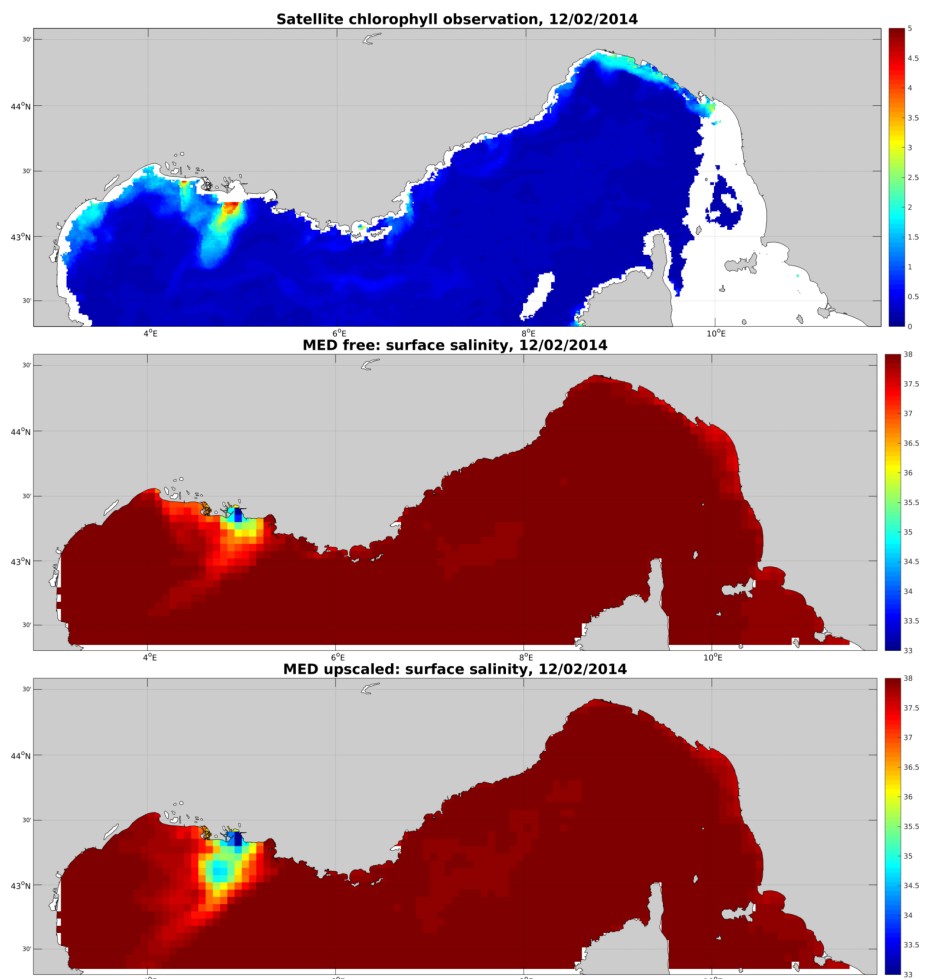

**Figure 9.** Comparison of the Rhône river plume on 12/Feb/2014 (upper panel) satellite chlorophyll image (middle panel) free parent model salinity (lower panel) and upscaled parent model salinity

solution. East and West of Corsica, the Corsican currents will reintroduce this water into the domain, and one can see how this repeated procedure will ultimately reduce discrepencies between parent and nested models.

**Deep temperature and salinity**

Some metrics considered above used surface salinity and temperature. However, upscaling modifies the 3D variables of tem-
5  perature and salinity.

Differences between the parent and the nested model temperature are locally important, e.g. on the bottom of the Gulf of Lions, or in the Eastern Corsican and Northern Current cores (with differences of up to 0.3°C). Similarly, the cores of both

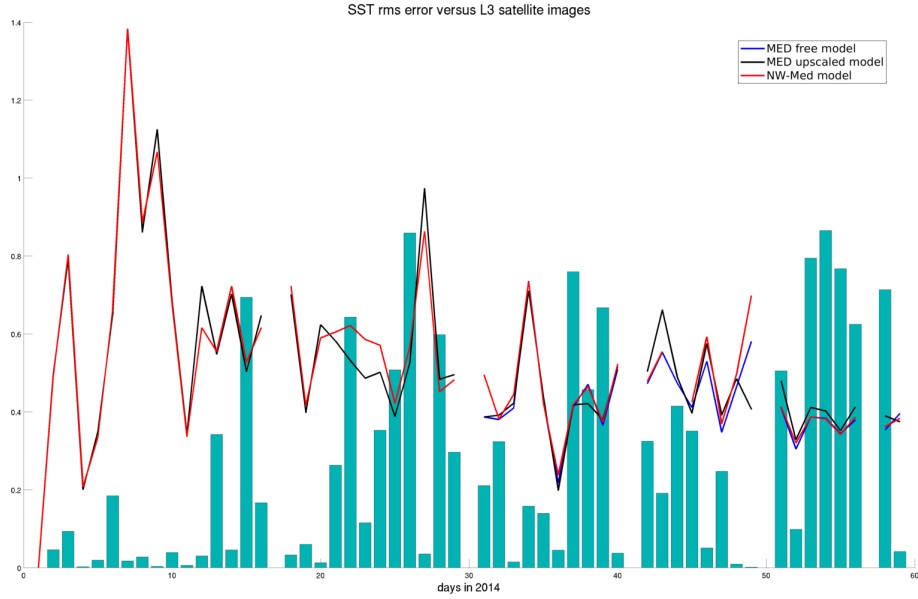

**Figure 10.** SST RMS error in the free model (black curve), nested model (red curve), upscaled model (blue curve) during the first 2 months of simulation. The bars represent the proportion of unclouded points in the L3 satellite image.

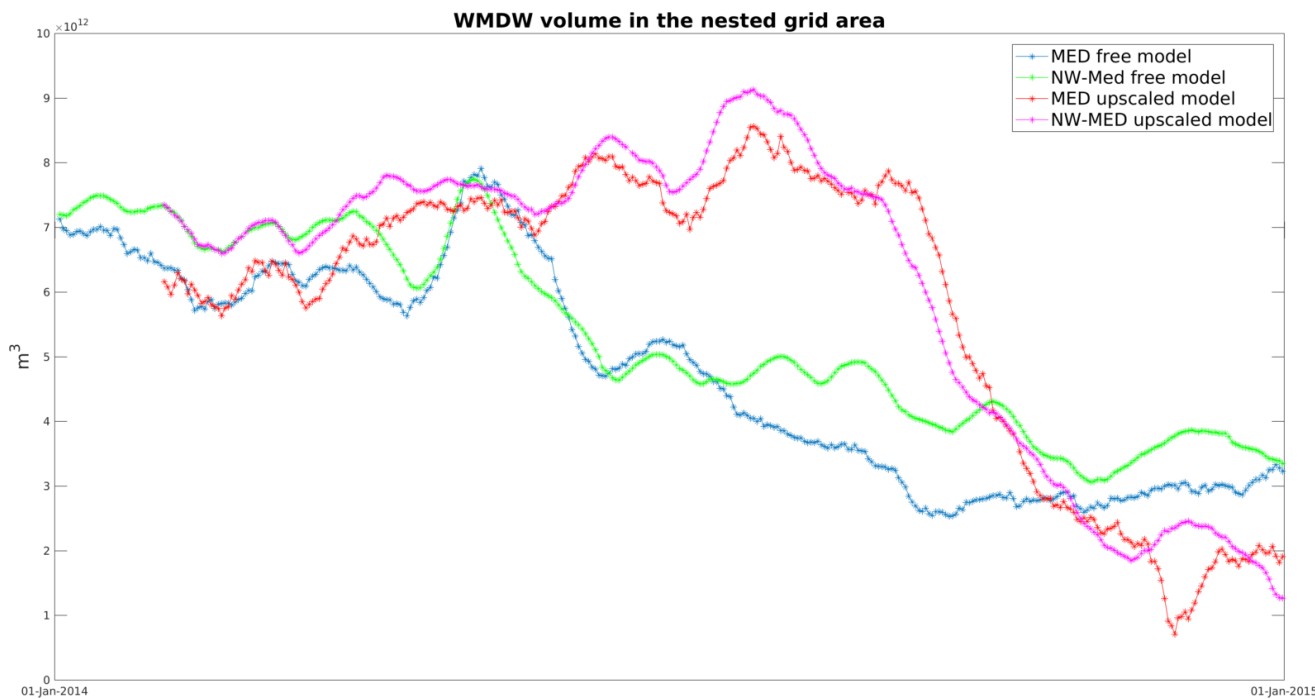

**Figure 11.** Time serie of total amount of WMDW in the area covered by the nested model: (blue) free parent model (green) nested model in the free model (red) parent model with upscaling (magenta) nested model in the upscaled model

| Metric | MED Free model | MED Upscaled model |
|---|---|---|
| Cross-shelf transport [m$^3$/s] | 99.6 10$^3$ | 85.2 10$^3$ |
| N.C. intensity [Sv] | 0.22 | 0.19 |
| Rhône Plume [km] | 95.1 | 88.0 |
| SST [°C] | 1.3 | 1.3 |
| WMDW [km$^3$] | 1563 | 1422 |

**Table 1.** Root Mean Square difference between parent and child model for the case of the free parent model and the upscaled parent model, for the defined metrics

Corsican Currents are saltier in the upscaled model, with differences of about 0.15 psu during the first assimilation cycle. For both temperature and salinity, upscaling is able to push the parent model toward the child model solution (not shown).

## 5 Conclusions

When a nested model is available, it usually benefits from higher resolution, and improved representation of some relevant
processes. However often, and particularly so in the operational oceanography context, there is no feedback from the nested model to the parent model. Data exchanges are limited to the parent model providing initial and/or boundary conditions to the nested model. Thus, the benefit of having a nested model is lost to the parent model.

The upscaling method consists in assimilating results from a sub-regional model into a regional (basin-wide) model, in order to emulate the feedback of two-way nesting. The underlying hypothesis is that the nested model is more realistic than the parent
model.

The method was tried out using a nested model configuration of the Mediterranean Sea and the North-Western basin, with a resolution ratio of 5. Data assimilation was performed using a localized ensemble Kalman Transform filter; as pseudo-observations, thinned 3D fields of temperature and salinity were used. The aim of this study is limited to verifying whether nesting feedback could be emulated by data assimilation; without trying to verify whether the nested model is indeed more
realistic than the parent one.

Whether upscaling was able to emulate two-way nesting, was measured using 5 metrics related to processes relevant in the study domain: the intensity of the Northern Current, the cross-shelf transport, the position of the Rhône river plume, sea surface temperature, and the quantity of Western Mediterranean Deep Water. These metrics show that the upscaling method is indeed able to emulate two-way nesting and bring the parent model closer to the child model. Only for sea surface temperature, the
RMS does not indicate an improvement, probably because this variable is essentially determined by atmospheric fluxes which are mostly identical (in our experiment) in the parent and child model. Some local improvements to sea surface temperature were observed, but are averaged out in the domain-wide RMS error.

By assimilating only temperature and salinity, velocity and transport metrics were also improved in the parent model. The ability to constrain the cross-shelf transport by T/S assimilation is also an indication that the data from a high-resolution glider

fleet would be beneficial to constrain the model. Finally, concerning the Rhône river plume, upscaling was able to strongly modify the plume direction when it was different in the parent and child models; the length of the plume was also modified. Qualitatively, when real chlorophyll observations were available, the nested and upscaled parent model seemed to be more realistic than the free parent model.

Advantages of using upscaling include the following. Most importantly of course, the parent model takes advantage of improvements in the nested model. In the current study, these improvements may be due to higher resolution, better representation of local processes, and the use of more realistic river discharges. In general, they may also have other causes, such as assimilation of local and/or very high resolution measurements (e.g. HF radar observations), atmospheric fields from a regional weather

forecasting model, or other more realistic boundary conditions. Another advantage is that over time, discrepencies between parent and nested model are attenuated. The parent model then provides more consistent boundary conditions to the nested model, and artefacts such as wave reflexion at the boundary may be avoided.

In the operational context, a supplementary advantage may appear. If a user is interested in a particular are not entirely covered by a nested model, it may be difficult for him to merge 2 products (the large-scale model, and the finer model not entirely

covering the area of interest). By default, the user may then use only the coarser model. If the nested model is upscaled into the large-scale model, this is the only product the user needs to consider.

The most important limitation of the method is that the child model should be more realistic than the parent model. Furthermore, the coupling with upscaling is not as strong as with real two-way nesting. Other limitations are linked to data assimilation

methods and are not different from the assimilation of real observations: (i) the data assimilation procedure itself uses approximations, and this could degrade the analysis; (ii) if the parent and child models are very different, the parent model could be unable to ingest the pseudo-observations. These limitations are investigated in the litterature in the context of assimilation of real observations, and potential solutions include (i) anamorphosis techniques (when a non-linear relation exists between model variables and observations), particle filters (when the error distribution cannot be considered Gaussian), etc; and (ii)

carefull specification of the observation error covariance matrix (and more specifically the contribution of the representativity error) to filter out processes of the nested model that cannot be represented in the parent model.

*Competing interests.* No competing interests are present.

*Acknowledgements.* This work has been carried out as part of the Copernicus Marine Environment Monitoring Service (CMEMS) UP-SCALING project. CMEMS is implemented by Mercator Ocean in the framework of a delegation agreement with the European Union.

Computational resources have been provided by the supercomputing facilities of the Consortium des Equipements de Calcul Intensif en Federation Wallonie-Bruxelles (CECI) funded by the Fond de la Recherche Scientifique de Belgique (FRS-FNRS).

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
