# Peer review of "Upscaling of a local model into a larger scale model"

_Ocean Science, 2018_

## Referee Comment (RC1) · Anonymous Referee #1 · 25 Sep 2018

General Comments:

In this study, the authors present the "upscaling" method in order to emulate two-way nesting, not implemented in operational systems. The "upscaling" technique consists of assimilating pseudo-observations from a child model in the parent model. The upscaling system is applied over the Mediterranean Sea, with the child model covering the north-western part of the region. The DA scheme is based on a variant of the EnKF assimilating only T, S. The concluding remark is that "upscaling" emulates successfully two-way nesting, but the study does not provide arguments whether the child model is more realistic than the parent.

The manuscript is clear, concise and well written. The study shows some interesting results and supports to some extent the argument of using the "upscaling" technique in

operational systems. However, my main concern is the limited impact of the study using a similar setup between the child and the parent models. Overall, I find the manuscript worthy of publication, after a minor revision. Please find below a list of comments that I would like the authors to address. The most important comments are listed first.

Specific comments:

1) My main concern is that "upscaling" appears to be feasible only if a similar setup is made between the child and the parent models. For instance, the authors use two models based on the same platform, i.e. NEMO, with an exact ration between horizontal grids and I suspect (not written in the text) with an identical vertical grid. All these are OK coinciding with the options to emulate two-way nested simulation. However, within a DA framework one would expect to see more general options, for instance, assimilating pseudo-observations on an entirely different grid (especially vertical for the T, S). The latter would support a more general argument for "upscaling" approaches, using for instance a different setup/grid/platform for the nested model. I leave it up to the authors choice if they wish to perform a DA experiment with a slightly different projection of the pseudo observations. However, I find useful the authors to discuss the limitations of their method.

2) page 2, line 16: "By upscaling the child model into the parent, the latter is brought closer to the former.". The benefits for the child model are obvious, though not so obvious for the parent model. Can the authors provide some guidance for "safe upscaling"? The way this work is constructed, suggests that a forecasting center should only "upscale" in case the child model has a similar modelling setup with the parent, e.g. same platform, vertical discretization, physics, parametrizations etc. The authors should also provide more information in the text about the setup of both models, in order to highlight their differences.

3) page 4, line 23: "these pseudo-observations coming from the nested model are considered independent". This is a very strong assumption, since observations are on

horizontal resolution 1/80°. In DA a common practice to avoid correlated errors is thinning or superobbing. Can the authors justify their option not to apply these techniques?

4) page 4, lines 21-22: "Ensemble Kalman filter" and page 5, line 11: "Ensemble Tranform Kalman Filter variant of the EnKF". Use also in page 4 the word "variant". In addition, the authors should write in this section the DA method in more details. For instance, it should be mentioned that this is a deterministic approach of the EnKF, i.e. pseudo-observations from the child model are not perturbed and the perturbation approach is only applied in the parent to obtain model errors. All these are not apparent to the reader, at least not before start reading the results section.

5) page 6, lines 13-14: "to update directly the tiles from the Mediterranean model restart files, influenced by the nested model, without including the other tiles in the state vector". This is an interesting technical capability of OAK, but if not mistaken that means that there is a crude correction cutoff in the neighboring tiles just outside the nested domain. I would assume that the localization is enough to constrain the correction in an area slightly broader than the nested domain. Can the authors clarify what is the purpose of this capability?

6) page 9, lines 7-8: "The ability to ... would be beneficial to constraint the model". This is more a concluding remark, rather than a result of the study. The phrase should be moved in the Conclusions section 5.

7) Figure 5. The SST is L4 or L3? In section 3.4 it is mentioned as L3.

8) Figure 8. The units are missing from the axes.

9) whole page 16: "Advantages of using upscaling include ...". This is a nice summary of "upscaling" advantages supporting the method. Can the authors provide possible disadvantages (if there are any) and suggest possible remedies?

Best regards.

---

## Referee Comment (RC2) · Anonymous Referee #1 · 24 Oct 2018

The authors have addressed all my comments in their response and i anticipate that the manuscript will be properly revised.

---

## Referee Comment (RC3) · Anonymous Referee #2 · 31 Oct 2018

The paper presents the upscaling technique applied to a realistic nested model configuration in the Mediterranean Sea domain. A NW-Med model is nested into a Mediterranean model (MED) with a downscaling factor of 5 and the aim is to prove that the upscaling technique is driving the parent model (MED) solution towards the child model one (NW-Med). The upscaling consists in assimilating the 3D temperature and salinity child model fields as pseudo-obs in the parent model. The upscaled model solution is thus closer to the child model when compared to the parent model using 5 different metrics.

General Comment

The paper presents the upscaling technique as a relevant scientific question in the operational model community, however it needs a lot of revisions to make it more a

scientific paper than a technical report. The English could be improved and detailed suggestions have been given but the reviewer is not mother tongue, thus take them carefully. The description of the methodology, results and conclusions appears some-time superficial and needs to be improved to be more complete and precise to allow their reproduction by fellow scientists. The figures must show always the same models (NW-Med, MED and upscaled). Labels, legends must be enlarged and captions im-proved. The RMSD could always be provided (MED-NW-Med and Upscaled-NWMed), and eventually be summarized in a table for the 5 metrics. The models' nomenclature should be consistent throughout the manuscript. Some references are missing. The underlying assumption that the child model has a better performance is only stated in the conclusions, while is should be clearly stated in the abstract or at least in the introduction, since this is not always true due to possible phase errors (in space and time) in the higher resolution models. Moreover the upscaling technique to be more powerful could weight pseudo-obs according to their misfit with real observations as-suring that the upscaling is stronger when and where the child model is closer to reality.

Please also note the supplement to this comment:
https://www.ocean-sci-discuss.net/os-2018-79/os-2018-79-RC3-supplement.pdf

**Supplement:**

**Upscaling of regional models into basin-wide models**

Luc Vandenbulcke and Alexander Barth

Research article in Special Issue: The Copernicus Marine Environment Monitoring Service (CMEMS): scientific advances

The paper presents the upscaling technique in a realistic configuration in the Mediterranean Sea domain. A NW-Med model is nested into a Mediterranean model (MED) with a downscaling factor of 5 and the aim is to prove that the upscaling technique is driving the parent model (MED) solution towards the child model one (NW-Med). The upscaling consists in assimilating the 3D temperature and salinity child model fields as pseudo-obs in the parent model. The upscaled model solution is thus closer to the child model when compared to the parent model using 5 different metrics.

**General Comment**

The paper presents the upscaling technique as a relevant scientific question in the operational model community, however it needs a lot of revisions to make it more a scientific paper than a technical report.

The English could be improved and detailed suggestions have been given but the reviewer is not mother tongue, thus take them carefully. The description of the methodology, results and conclusions appears sometime superficial and needs to be improved to be more complete and precise to allow their reproduction by fellow scientists. The figures must show always the same models (NW-Med, MED and upscaled). Labels, legends must be enlarged and captions improved. The RMSD could always be provided (MED-NW-Med and Upscaled-NWMed), and eventually be summarized in a table for the 5 metrics. The models' nomenclature should be consistent throughout the manuscript. Some references are missing.

The underlying assumption that the child model has a better performance is only stated in the conclusions, while is should be clearly stated in the abstract or at least in the introduction, since this is not always true due to possible phase errors (in space and time) in the higher resolution models. Moreover the upscaling technique to be more powerful could weight pseudo-obs according to their misfit with real observations assuring that the upscaling is stronger when and where the child model is closer to reality.

I recommend to accept the paper for further publication after a major revision.

**Specific Comments**

**Abstract**
Line 5: *Therfore* instead of therefor and I would take out "*in practice*"
Line 6: …" to replace the missing model feedback…" I would insert "*child model or high resolution model*".
Line 10: I suggest to rephrase something like:
"A basin scale model simulation is compared to one simulation…, and another model analysis which applies the upscaling technique…"

**Introduction**
Line 15: "reanalyses, analyses and forecasts"
Line 16: " …by different institutes within the regional monitoring and forecasting centers.."
Line 24: could you insert a reference for this?
Line 6  page 2: I would take out "in this artice"

Line 12 page 2: I would substitute *basin-scale* with *regional*
Line 13 page 2: "…in the basin-scale model, … is to obtain"
Line 14 page 2: "(along with…)"
Please rephrase the entire sentence, it seems too informal to me
Line 16 page 2: I suggest *"...to the child model will progressively gain consistency with the child model solution within its domain, being beneficial for the child model over time.*"
Line 20 page 2: do you have references for this?
Line 21 page 2: Is this pertinent? I do not see the connection, please explain.
Line 26 page 2: Other re-initialization techniques have been used blending, through optimal interpolation, coarse resolution operational analyses and coastal observations in so called Rapid Environmental Assessment experiments. Please give a look at *Simoncelli et al.(2011)*, they show improvements in the nested coastal model performance using observations.
Line 35 page 2: I am not sure that the syntax is correct please check the English.
Line 3 page 3: "Therefore acknowledging that operational…, Schulz-Stellenfleth and Stanev (2016) strongly…"

**2.1 Hydrodynamic Model**
Line 13: I would substitute *tried out* with *has been implemented*
Line 15: *"created by the junction of the Eastern and Western Corsican Currents"* I would cite *Pinardi et al (2015)*
*Line 17:* I would add some recent references *Pinardi et al (2015), Somot et al. (2016), Simoncelli and Pinardi in von Schuckman et al. (2018).*
Line 21: The resolution of MFS is $1/24^{th}$ (Clementi et al. 2017, https://doi.org/10.25423/cmcc/medsea_analysis_forecast_phy_006_013) of a degree since October 2017. The reanalysis (https://doi.org/10.25423/medsea_reanalysis_phys_006_004 Simoncelli et al., 2014, 2016) is still at $1/16^{th}$. Please check the CMEMS product catalogue for the correct acknowledgment. The system at $1/16^{th}$ (https://doi.org/10.25423/medsea_analysis_forecast_phys_006_001)is still operational at INGV http://medforecast.bo.ingv.it/mfs-copernicus/
Line 30: please specify for reproducibility issues which analyses has been used, I guess the 1/16? Or the reanalyses? Please clarify and insert the reference.
Line 32: ERA Interim is not at a resolution of $1/8^{th}$ of degree, it is 0.75 degrees!!! You might have re interpolated it from $1/8^{th}$ to $1/16^{th}$. Please explain it and add the Dee et al () reference of ERA Interim.

Line 1 page 4: Which literature?
Line 2: 5 rivers' data comes from?
Line 4: Please consider the CMEMS has 39 rivers, thus it is not very much coherent. Please look at the CMEMS products descriptions and may be cite the reanalysis instead. Rivers are described in the Simoncelli et al 2016.

Figure 1: Please increase the axis font, not readable now. I would show the two models' salinity fields to emphasize the differences due to the daily river outflow, instead of the difference. In fact, you describe the different plumes in the manuscript.

Line 10: please describe more in detail Fig1b eventually, isn't it the difference among the two models' salinity after 1 month of simulation? Please improve also the caption.

**2.2**
Line 16: I would re-phrase something like "*In order to assimilate* …, different set ups could be implemented (adopted, applied) depending on …"

Line 21: please improve the description of the settings, it contains repetitions.
Line 23-25: Start a new phrase please and please say something more about the statement that the T and S pseudo-obs are considered independent. You mean that you assume that even if it is not the case. You assimilate the full resolution 3D T and S fields? None thinning? Please motivate a bit this part. The word *also* could be neglected and substitute ";".
It looks like you wrote this in a rush without much care.

Line 31: random not randon

Line 7 Page 5: Why did you select 1 month of spin up time? Please start a new phrase and integrate a bit on that. What you use in the evaluation is the ensemble mean of the 100 members?

Line 1 Page 6: it is not clear to me "*...and its addition observation localization*", could you please explain?
Line 7: many are the sensitivity experiments with different observation errors, maybe you could insert a table.

**3. Metrics**

Line 17: Please add something about: .. these metrics have been computed to compare model solutions. What do you want to show? Upscaling model solution (is it an ensemble mean of the 100 members?) with the basin scale simulation?

**3.1**
Line 23: see previous comment. What do you want to show? Does this metric tell you if the upscaling procedure is driving the upscaled model solution towards the NW-Med? Please integrate a bit.

**3.2**

Line 28: again, see previous comments. I would integrate at the beginning of the section, and in each sub-section I would state what exactly is expected from each metrics.

**3.3**
I do not really understand what you are going to present in the results. Plume length/direction from the free and upscaled models? The comparison model-obs using sat chlla images? "Furthermore,…" What does it mean you do it qualitatively, quantitatively, both?

**3.4**
ok

**3.5**
Line 15: What do you mean bay "depth is larger than 1000m"? Is it the depth reached by deep convection? Maybe you want to put it at the end of the first phrase at line 13. Here you are describing the WMDW characteristics.
Line 17: Which tail? How could the river see that? Please explain or describe with some detail.\

**4. Results**
Figure 3: please enlarge the font.
Line 5: The difference of what?
Line 9: Isn't it Fig.1b the difference between unperturbed parent and child model in salinity after 1 month of spin up?
Line 10: Why should I trust the child model more than the parent? You did not provide any model performance. It is not easy to improve significantly the smooth solution of the coarse parent model (phase errors are common).

Figure 4: please include a line to indicate the section location on the map.

Line 16: The scope of Figure 5 is to show that the upscaling technique is bringing the upscaled model close to the NW-Med one. RMS difference could be provided towards the satellite SST to show this. However, this rise the question: " why don't you consider to weight the assimilation of pseudo observations according to the misfit with the observed SST, giving more weight where the pseudo-obs are loser to sat obs? Please keep the same notation to call the models (MED, NW-Med…).

Please switch 4.1 and 4.2 (thus fig. 6 and 7) to be consistent with 3.1 and 3.2.

**4.1**

Line 21: For which models?

Line 2 Page 9: Why the NW-Med model is not shown in Figure 6? Please include it for consistency in presenting the results, since you want to prove that the upscaled model is driven towards the NW-Med solution. Again please keep the same nomenclature to facilitate the reader. Moreover RMSD could be computed as in 4.2 case.

Lines 9-12: What is this meaning? Is this correct? Is this coherent with NW-Med?

Figure 6: please increase the font in the legend and include NW-Med.

**4.2**

Figure 7: Please insert NW-Med and the legend.

Lines 4 -6Page 11: I suggest to rephrase *"...this metrics cannot **be** used to compare and validate the models since observations are not available to compute the real NC transport. However, it shows that upscaling of.."*

**4.3**

As for the other metrics I would include the NW-Med to show its consistency with the upscaled model. At Line 15 you mention the nested (NW-Med) model but it is not shown. RMSD could be computed again as more robust argument of your results.
Line 17: a quantitative estimate of model performance can be provided again by RMSD of MED, and upscaled model towards NW-Med.

Figure 9: This figure presents MED and NW-Med, why not the upscaled model? I suggest to shaw the three salinity fields.

Line 3 Page 12: Is the increase in salinity observed, is it consistent with observations or only with NW-Med? I would include validation with observations, since a lot of them should be available by REP in situ observations fromCMEMS.

**4.4**

Figure 10 should be mentioned since the beginning of the paragraph to help the reader. Please harmonize the models' nomenclature in the text but also in the legend and caption of figure 10. The validation (actually the only one that is provided with observations) should cover the entire 2014 year and not only two months. The mean RMSE of the three models should be provided as well to support your results. Upscaling should consider the performance of the child model and assimilate only, or give more weight, in those parts of the domain where the child model is close to observations. Without considering models' performance upscaling could force the parent model towards a wrong solution.

Lines 6-8 Page 13: why not? REP in situ temperature and salinity profiles are available from CMEMS. I encourage to look at them and compute some validation to make your paper more robust.

Line 8: "Upscaling is able to bring the differences back to  the parent model" or the child model?

**4.5**

Here you mention also the child-model of the upscaled model and you speculate without any proof about the largest consistency in the nested upscaled models. I recommend to include a figure to show it. Or if you want to use Fig. 11 please give the RMSD of the model couples or include a specific comment to the figure (Line 7-8?)

Line 10: Therefore

**5. Conclusions**

Lines 4-9 Page 15: "The underlining hypothesis…" You only state it in the conclusions while it should be written clearly both in the abstract and the introduction. In this way the reader is aware

that you only aim to relax the child model solution to the parent model solution, independently from the model performances.

Line 13: please use child instead of nested.

Line 15: Please be more precise and refer to the figures when this is shown.

Lines 1-6 Page 16: This general statement might confuse the reader since your child models never assimilate observations. Please rephrase to underline this.

Line 6: please include a reference since you do not show this in the paper.

Line 7-end: The advantage might be real is the upscaled solution has a better skill towards observations. In the last phrase: If the high resolution model is upscaled into the basin-scale one? Please notice that in CMEMS regional systems are the basin-scale ones.

---

## Author Comment (AC2) · 27 Nov 2018

We thank the reviewer for his work

---

## Author Response (AR1)

Dear Madam, Sir,

We thank 2 anonymous reviewers for the precise and helpful comments about our manuscript. We realize that a lot of time went into the reviewing process, giving the precise (and justified) remarks. We answered each of the comments, and modified the text and the figures accordingly. The revised version of the manuscript is much clearer, some mistakes was corrected, some new references were added, and some more links to CMEMS products were given, all according to the reviewer's comments.

In the replies to the reviewer #1, we announced to add a new annex with details of the data assimilation method. After completing also the correction of the manuscript for reviewer #2, we think that these details should be provided directly in the concerned section (instead of creating an annex). This is the only change in the reviewed paper, inconsistent with what we replied in the OSD forum. It is written in red in the text below.

We think the revised paper is much better than the original one, and submit it to you for consideration for publication in OS.

The replies to the reviewers were posted in the OSD discussion. They are copied below for your convience.

Reviewer 1
Specific comments:
1) My main concern is that "upscaling" appears to be feasible only if a similar setup is made between the child and the parent models. For instance, the authors use two models based on the same platform, i.e. NEMO, with an exact ration between horizontal grids and I suspect (not written in the text) with an identical vertical grid. All these are OK coinciding with the options to emulate two-way nested simulation. However, within a DA framework one would expect to see more general options, for instance, assimilating pseudo-observations on an entirely different grid (especially vertical for the T, S). The latter would support a more general argument for "upscaling" approaches, using for instance a different setup/grid/platform for the nested model. I leave it up to the authors choice if they wish to perform a DA experiment with a slightly different projection of the pseudo observations. However, I find useful the authors to discuss the limitations of their method.
The reviewer is entirely right that the configuration used to test the upscaling method is based on a nested grid setup using the same model code (Nemo) for both the parent and child grid; and furthermore the vertical grid is also identical (only the horizontal grid is different). This may influence the conclusion compared to a configuration with 2 different model codes. However we think that it is not a fundamental limit of the method a) Concerning the vertical grid, in the "normal" case of assimilating real observations, the latter are on a different grid than the model. Similarly if the child model was on a different vertical grid than the parent model, it would still contain useful information, and be worth to be assimilated in the parent model. What may happen however, is that some observations could be lost (e.g. the lowest model of the child model could be out-of-grid in the parent model)
b) if different model codes are used, the models could represent different processes. Hence, this should be taken into account by modifying the (representativity part of the) observation error covariance matrix. Examples of contributions to the representativity error could be
- different vertical coordinates
- different representations of the surface: rigid lid, free surface (with a linear or non-linear representation e.g. in Nemo)
- hydrostatic model, or not

- different atmospheric forcing fields
- different turbulent closure schemes
- different numerical schemes for advection, horizontal diffusion etc.
It is our opinion however, that between the parent and child models, the most striking difference is the horizontal resolution, and that therefor, the general conclusions of the paper are valid, and upscaling should not be limited to the case of parent and child models being identical. This is now better explained in the paper

2) page 2, line 16: "By upscaling the child model into the parent, the latter is brought closer to the former.". The benefits for the child model are obvious, though not so obvious for the parent model. Can the authors provide some guidance for "safe upscaling"? The way this work is constructed, suggests that a forecasting center should only "upscale" in case the child model has a similar modelling setup with the parent, e.g. same platform, vertical discretization, physics, parametrizations etc. The authors should also provide more information in the text about the setup of both models, in order to highlight their differences.

If one considers that the child model is better in its domain than the parent model (e.g. by comparison with real observations), then it would be desirable to upscale it into the parent model. This would be the case is some processes are dependent on resolution, in straits, etc; and is closely linked to the first specific comment in the review. We provide now a table in the annex of the paper giving details about the setup of both models; but upscaling should not be limited to identical parent and child models (see answer to comment 1)

3) page 4, line 23: "these pseudo-observations coming from the nested model are considered independent". This is a very strong assumption, since observations are on C2 OSD Interactive comment Printer-friendly version Discussion paper horizontal resolution 1/80◦ . In DA a common practice to avoid correlated errors is thinning or superobbing. Can the authors justify their option not to apply these techniques?

The reviewer is correct, that the assumption of spatially independant pseudo-observations is very strong. We are actually working on a non-diagonal observation error covariance matrix, but this is a large work that would not fit into the current paper. However, the assumption is partly alleviated by increasing the (diagonal) part of the matrix, in order to compensate for the (missing) non-diagonal elements. Increasing the diagonal elements in the matrix by a factor 3, for example, is similar to thinning observations with a factor 3. This is now stated in the paper.

4) page 4, lines 21-22: "Ensemble Kalman filter" and page 5, line 11: "Ensemble Tranform Kalman Filter variant of the EnKF". Use also in page 4 the word "variant". In addition, the authors should write in this section the DA method in more details. For instance, it should be mentioned that this is a deterministic approach of the EnKF, i.e. pseudo-observations from the child model are not perturbed and the perturbation approach is only applied in the parent to obtain model errors. All these are not apparent to the reader, at least not before start reading the results section.

We added an annex to the paper with the details of the data assimilation filter used in the study.
In the final revised version of the paper, we decided not to enter this information in an annex, but directly in the paper, as suggested by the reviewer.

5) page 6, lines 13-14: "to update directly the tiles from the Mediterranean model restart files, influenced by the nested model, without including the other tiles in the state vector". This is an interesting technical capability of OAK, but if not mistaken that means that there is a crude correction cutoff in the neighboring tiles just outside the nested domain. I would assume that the localization is

enough to constrain the correction in an area slightly broader than the nested domain. Can the authors clarify what is the purpose of this capability?

In the state vector of the parallelized parent model, we include the tiles covered by the nested model, but also the tiles immediately around that area. Therefore, as the reviewer correctly assumes, the correction is not cut off at the margins of the area covered by the nested model, but propagates outside. The extent of the correction outside the area, depends on the radius used in the localization method. This is now better explained in the paper.

6) page 9, lines 7-8: "The ability to ... would be beneficial to constraint the model". This is more a concluding remark, rather than a result of the study. The phrase should be moved in the Conclusions section 5.

We moved the remark to the conclusions

7) Figure 5. The SST is L4 or L3? In section 3.4 it is mentioned as L3.

In the study (section 3.4), L3 satellite images are used. Only in figure 5 is the L4 image used for visual comparison of model and satellite image. This is now clarified in the article.

8) Figure 8. The units are missing from the axes.

added units

9) whole page 16: "Advantages of using upscaling include ...". This is a nice summary of "upscaling" advantages supporting the method. Can the authors provide possible disadvantages (if there are any) and suggest possible remedies?

The reviewer is right that the list of advantages should be accompagnied with a list of possible limitations (or disadvantages). This is now included in the article, and copied here:

a) the child model should be "better" than the parent model

b) exactly as when assimilating real observations, the data assimilation procedure itself uses approximations, and this could degrade the analysis

c) if the parent and child models are very different, the parent model could not manage to ingest the pseudo-observations

d) the coupling with upscaling is not as strong as with real two-way nesting

Potential remedies for limitations b and c would be

b) see all the research about this problem (in the context of assimilation of real observations), such as anamorphosis techniques (when a non-linear relation exists between model variables and observations), particle filters (when the error distribution cannot be considered Gaussian), etc

c) the observation error covariance matrix should be specified carefully to filter out the processes of the child model, that cannot be represented in the parent model

Reviewer 2

**Abstract**

Line 5: *Therfore* instead of therefor and I would take out "*in practice*"
Changed to Therefore

Line 6: …" to replace the missing model feedback…" I would insert "*child model or high resolution model*". Done

Line 10: I suggest to rephrase something like:
"A basin scale model simulation is compared to one simulation…, and another model analysis which applies the upscaling technique…"
Changed

**Introduction**

Line 15: "reanalyses, analyses and forecasts" Changed
Line 16: " …by different institutes within the regional monitoring and forecasting centers.." Changed
Line 24: could you insert a reference for this? We rephrased the sentence instead
Line 6  page 2: I would take out "in this artice" Removed

Line 12 page 2: I would substitute *basin-scale* with *regional* Changed

Line 13 page 2: "…in the basin-scale model, … is to obtain" Changed
Line 14 page 2: "(along with…)" Changed
Please rephrase the entire sentence, it seems too informal to me Rephrased
Line 16 page 2: I suggest "*…to the child model will progressively gain consistency with the child model solution within its domain, being beneficial for the child model over time*." Changed
Line 20 page 2: do you have references for this?  Added Mason et al 2010, Debreu et al 2012
Line 21 page 2: Is this pertinent? I do not see the connection, please explain. If we assimilate pseudo-observations coming from a nested model, and this improves our (parent) model, then the child model is a pseudo-measurement device which can be seen as a replacement for (costly) real measurement devices.
Line 26 page 2: Other re-initialization techniques have been used blending, through optimal interpolation, coarse resolution operational analyses and coastal observations in so called Rapid Environmental Assessment experiments. Please give a look at *Simoncelli et al.(2011)*, they show improvements in the nested coastal model performance using observations. Added
Line 35 page 2: I am not sure that the syntax is correct please check the English. Changed
Line 3 page 3: "Therefore acknowledging that operational…, Schulz-Stellenfleth and Stanev (2016) strongly…" Changed

**2.1 Hydrodynamic Model**

Line 13: I would substitute *tried out* with *has been implemented* Changed
Line 15: "*created by the junction of the Eastern and Western Corsican Currents*" I would cite *Pinardi et al (2015)* Added
*Line 17*: I would add some recent references *Pinardi et al (2015), Somot et al. (2016), Simoncelli and Pinardi in von Schuckman et al. (2018).* Added
Line 21: The resolution of MFS is $1/24^{th}$ (Clementi et al. 2017, https://doi.org/10.25423/cmcc/medsea_analysis_forecast_phy_006_013) of a degree since October 2017. The reanalysis (https://doi.org/10.25423/medsea_reanalysis_phys_006_004 Simoncelli et al., 2014, 2016) is still at $1/16^{th}$. Changed

Line 30: please specify for reproducibility issues which analyses has been used, I guess the 1/16? Or the reanalyses? Please clarify and insert the reference. Added

Line 32: ERA Interim is not at a resolution of $1/8^{th}$ of degree, it is 0.75 degrees!!! You might have re interpolated it from $1/8^{th}$ to $1/16^{th}$. Please explain it and add the Dee et al () reference of ERA Interim. Modified and added reference

Line 1 page 4: Which literature? Removed « from the litterature »
Line 2: 5 rivers' data comes from? Various regional websites, not added in the article
Line 4: Please consider the CMEMS has 39 rivers, thus it is not very much coherent. Please look at the CMEMS products descriptions and may be cite the reanalysis instead. Rivers are described in the Simoncelli et al 2016. Added in the text that CMEMS has many more rivers

Figure 1: Please increase the axis font, not readable now. I would show the two models' salinity fields to emphasize the differences due to the daily river outflow, instead of the difference. In fact, you describe the different plumes in the manuscript.
We tried this out, but due tu large salinity variability, a side-by-side plot of salinity does not show clearly the salinity difference. Therefore we chose to show directly the difference

Line 10: please describe more in detail Fig1b eventually, isn't it the difference among the two models' salinity after 1 month of simulation? Please improve also the caption.
Yes Fig 1b is the difference of (nested) model salinity after 1 month, when using climatological Rhone discharge or real, daily discharge. This is now written more clearly in the caption

**2.2**
Line 16: I would re-phrase something like "*In order to assimilate …,* different set ups could be implemented (adopted, applied) depending on …" Changed

Line 21: please improve the description of the settings, it contains repetitions.
Line 23-25: Start a new phrase please and please say something more about the statement that the T and S pseudo-obs are considered independent. You mean that you assume that even if it is not the case. You assimilate the full resolution 3D T and S fields? None thinning? Please motivate a bit this part. The word *also* could be neglected and substitute ";". It looks like you wrote this in a rush without much care. Improved lines 21-25 as suggested
As the reviewer correctly supposed, we do perform thinning (in the horizontal), and this is now also written in the text. We apologize for forgetting to write it in the article first submitted

Line 31: random not randon Changed

Line 7 Page 5: Why did you select 1 month of spin up time? Please start a new phrase and integrate a bit on that. What you use in the evaluation is the ensemble mean of the 100 members?
In a previous (single) model spin-up, we noticed that the kinetic energy reaches a more-or-less stable value in just a few days, therefor a spin-up of 1 month was considered sufficient. The same is supposed for the ensemble members. As we are now running an ensemble of 100 members, longer spin-up would translate into large computational cost.

Line 1 Page 6: it is not clear to me "*...and its addition observation localization*", could you please explain?

It is explained just afterwards (lines 2-3. In  means that the model domain is cut into subdomains (water columns in our case) where the analysis is performed separately. Moreover,  in each subdomain, only relevant observations are considered (i.e. observations that are far away and will have no impact, are not considered during the computation).

Line 7: many are the sensitivity experiments with different observation errors, maybe you could insert a table.
We agree that there are 4 experiments (each with a value for temperature and salinity observation errors), but these all fit into 1 line in the text. Maybe it is easier to keep them in the text ?

**3. Metrics**

Line 17: Please add something about: .. these metrics have been computed to compare model solutions. What do you want to show? Upscaling model solution (is it an ensemble mean of the 100 members?) with the basin scale simulation?
Added some explation to Line 17.  About the « upscaled »  model, we keep one unperturbed ensemble member. The other 99 members of the ensemble are perturbed and are used only to  create the model error space. All 100 members are updated daily by the data assimilation procedure.

**3.1**
Line 23: see previous comment. What do you want to show? Does this metric tell you if the upscaling procedure is driving the upscaled model solution towards the NW-Med? Please integrate a bit.
Yes exactly, we want to see if the upscaling procedure pushes the parent model towards the NW-Med model. This is now explained better at line 23.

**3.2**
Line 28: again, see previous comments. I would integrate at the beginning of the section, and in each sub-section I would state what exactly is expected from each metrics.

**3.3**
I do not really understand what you are going to present in the results. Plume length/direction from the free and upscaled models? The comparison model-obs using sat chlla images? "Furthermore,…" What does it mean you do it qualitatively, quantitatively, both?
Yes we compare plume length and directions in between models, and also between model and real observations. This is different compared to the previous metrics where we did not have real measurements. This is now better explained in the text.

**3.4**
ok

**3.5**
Line 15: What do you mean bay "depth is larger than 1000m"? Is it the depth reached by deep convection? Maybe you want to put it at the end of the first phrase at line 13. Here you are describing the WMDW characteristics.
Following Bosse at al (2015), we consider that water is WMDW if it is within a certain T and S range, AND if it is situated below 1000m depth

Line 17: Which tail? How could the river see that? Please explain or describe with some detail.\
Bosse et al (2015) show the T-S diagram and indicated the water masses (left plot below). We reproduce this plot from model results (right plot below) and obtain a longer tail (it goes to ~12.7°C and slightly lower than 38.45 psu).The definition from Bosse at al (2015) would result only in the red rectangle in the right plot below. Therefore we slightly adapt the WMDW definition in order to capture the whole tail. We decided not to include these 2 figures in order to keep the paper more consise.

[Figure]

[Figure]

**4. Results**
Figure 3: please enlarge the font. Changed
Line 5: The difference of what? Temperature, added in the text and in figure caption
Line 9: Isn't it Fig.1b the difference between unperturbed parent and child model in salinity after 1 month of spin up? Fig 1b is the salinity difference of the nested model, when forced with climatological or with real Rhone river discharge data. Figure 3 is the difference between parent and nested model (for temperature), and the equivalent for salinity is not shown in the article
Line 10: Why should I trust the child model more than the parent? You did not provide any model performance. It is not easy to improve significantly the smooth solution of the coarse parent model (phase errors are common).
Also following recommendation from reviewer 1, we now stated clearly that we *suppose* from the start that the nested model is better (in some sense) than the parent. We do not try to prove the hypothesis is actually valid, and sometimes, it could be wrong. Exactly as the reviewer points out, when the nested model represents small scales that are actually out of phase with reality, it could have higher RMS errors than the parent model that does not represent small scales at all.

But supposing that the nested model is « better », our objective with upscaling is to bring the parent model closer to the nested model (i.e. emulate nesting feedback). This is now written in the introduction.

Figure 4: please include a line to indicate the section location on the map. Done

Line 16: The scope of Figure 5 is to show that the upscaling technique is bringing the upscaled model close to the NW-Med one. RMS difference could be provided towards the satellite SST to show this. However, this rise the question: " why don't you consider to weight the assimilation of pseudo observations according to the misfit with the observed SST, giving more weight where the pseudo-obs are loser to sat obs? Please keep the same notation to call the models (MED, NW-Med…).
This is an interesting suggestion but has also the following limitation. If we have many observations to validate the nested model, then we could as well assimilate these (real) observations into the parent model. Upscaling is interesting mostly when there are few or no real observations.
In the case of SST however, it is a very interesting suggestion. When the nested model is well-validated by SST, maybe we can trust it more also for other variables and assimilate those in the parent model to complement the (real) SST observations. This could be the topic of a follow-up study.

Please switch 4.1 and 4.2 (thus fig. 6 and 7) to be consistent with 3.1 and 3.2. Done, we rather switched 3.1 and 3.2

**4.1**

Line 21: For which models? Parent model, free and upscaled. Added in the text

Line 2 Page 9: Why the NW-Med model is not shown in Figure 6? Please include it for consistency in presenting the results, since you want to prove that the upscaled model is driven towards the NW-Med solution. Again please keep the same nomenclature to facilitate the reader. Moreover RMSD could be computed as in 4.2 case.
Please see the answer to comment 4.2 (Fig. 7)

Lines 9-12: What is this meaning? Is this correct? Is this coherent with NW-Med?
Yes it can be seen from the figure that both lines are closer one-to-the-other after the first month. There is also a temporal delay in between the 2 curves in August-September. The text merely describes this.

Figure 6: please increase the font in the legend and include NW-Med.
We will modify the figure fonts

**4.2**

Figure 7: Please insert NW-Med and the legend.
Once upscaling modifies the parent model, the child model gets different open-sea boundary conditions from the parent, and starts to be different as well. Thus, for Figures 6 and 7, we could add 2 NW-Med curves to the figures (one corresponding to the free run and one corresponding to the upscaling run). For the Northern Current intensity, it gives this :

[Figure]

This figure becomes difficult to interpret. There is no general trend, like for example a systematically more intense Northern Current in nested models. However, what's interesting is that generally speaking, compared to the free Med model (blue curve), the yellow curve seems to get closer to the purple (child model of the upscaled Med model). The discrepancy between yellow and purple is smaller than the discrepancy between blue and red. This can be verified by looking at RMSD errors between parent and child model in the two cases (free model or upscaled model), respectively 0.22 and 0.19 Sv. The RMSD for all metrics have now been included in a table as suggested by the reviewer.

Lines 4 -6Page 11: I suggest to rephrase *"…this metrics cannot **be** used to compare and validate the models since observations are not available to compute the real NC transport. However, it shows that upscaling of.."*
But the metric *can* be used to inter-compare models, so it might be confusing to say « this metric cannot be used to compare … ». Therefore we rephrased as :
« For the purpose of our study, this metric cannot be used to validate the model since real measurements of the Northern Current transport are not available; but (as for the previous metric), it can be used to compare models, … »

**4.3**
As for the other metrics I would include the NW-Med to show its consistency with the upscaled model. At Line 15 you mention the nested (NW-Med) model but it is not shown. RMSD could be computed again as more robust argument of your results.
On Fig. 8, the arrows for the upscaled and (not-shown) NW-MED models superpose and are indistinguishable
Line 17: a quantitative estimate of model performance can be provided again by RMSD of MED, and upscaled model towards NW-Med.
We have computed the RMSD and included it in a table

Figure 9: This figure presents MED and NW-Med, why not the upscaled model? I suggest to shaw the three salinity fields.
It is indistinguishable from the nested model

Line 3 Page 12: Is the increase in salinity observed, is it consistent with observations or only with NW-Med? I would include validation with observations, since a lot of them should be available by REP in situ observations fromCMEMS.

We agree that it could be checked whether it is realistic or not. However even if it is *not* realistic, if the nested model predicts it, and upscaling can make the parend model predict it as well, then upscaling is doing what we hoped. Our hypothesis is always that the nested model is « better ».
In other words, if nesting predicts a saltier Corsican Current core, but it is actually not true, what do we hope from upscaling ? That the parent model also increases salinity in the core, or not ?
For this reason, we actually did not check if the nested model is more realistic or not. There are lots of other papers showing the impact of nesting.

**4.4**

Figure 10 should be mentioned since the beginning of the paragraph to help the reader.
Done
Please harmonize the models' nomenclature in the text but also in the legend and caption of figure 10. The validation (actually the only one that is provided with observations) should cover the entire 2014 year and not only two months.

In the text is written : « A similar plot for the whole of 2014 shows that the situation worsens during summer (errors of 3 ◦ C) both for parent and child model; the difference in between models is hidden by the temporal variability of the error (not shown). »
If we plot the entire year, due to the scale going to 3°C instead of 1.4°C, we would almost not distinguish the red, black and blue curves anymore at all.

 The mean RMSE of the three models should be provided as well to support your results. Upscaling should consider the performance of the child model and assimilate only, or give more weight, in those parts of the domain where the child model is close to observations. Without considering models' performance upscaling could force the parent model towards a wrong solution.

This comment is related to a suggestion higher in the review. We agree that this could be a way forward for SST, but maybe in a follow-up study. Apart from SST, for the other metrics, there are no (real) data to choose whether or not to trust the nested model. The hypothesis of this study is that the nested model **is** better, and the objective is to bring the parent model closer to the nested model.

Lines 6-8 Page 13: why not? REP in situ temperature and salinity profiles are available from CMEMS. I encourage to look at them and compute some validation to make your paper more robust.

This comment is again related to the previous one, and other ones before. In this study, we **only** aim at bringing the parent model closer to the nested model. Validating the nested model, such as doing a QUID for it, is out of our scope (and a whole lot of work).
On this occasion of in-depth temperature, we remembered the reader of the hypothesis and objective of the paper.

Line 8: "Upscaling is able to bring the differences back to  the parent model" or the child model?
We rephrased this so that is it more clear

**4.5**

Here you mention also the child-model of the upscaled model and you speculate without any proof about the largest consistency in the nested upscaled models. I recommend to include a figure to show it. Or if you want to use Fig. 11 please give the RMSD of the model couples or include a specific comment to the figure (Line 7-8?)
We have added a table with the RMSD of model couples, as suggested by the reviewer

Line 10: Therefore Changed

**5. Conclusions**

Lines 4-9 Page 15: "The underlining hypothesis…" You only state it in the conclusions while it should be written clearly both in the abstract and the introduction. In this way the reader is aware that you only aim to relax the child model solution to the parent model solution, independently from the model performances.
Yes exactly. Sorry for this. It is now written in the introduction

Line 13: please use child instead of nested. changed

Line 15: Please be more precise and refer to the figures when this is shown. changed

Lines 1-6 Page 16: This general statement might confuse the reader since your child models never assimilate observations. Please rephrase to underline this. Rephrased

Line 6: please include a reference since you do not show this in the paper. Changed the text

Line 7-end: The advantage might be real is the upscaled solution has a better skill towards observations. In the last phrase: If the high resolution model is upscaled into the basin-scale one? Please notice that in CMEMS regional systems are the basin-scale ones.
We added explicitely in the text the potential disadvantages of the method, which include the fact that the nested model may be worse than the parent.

[revised manuscript text omitted]

---

## Referee Report (RR1)

**Upscaling of regional models into basin-wide models**

**Luc Vandenbulcke and Alexander Barth**

Research article in Special Issue: The Copernicus Marine Environment Monitoring Service (CMEMS): scientific advances

The paper presents the upscaling technique in a realistic configuration in the Mediterranean Sea domain. A NW-Med model is nested into a Mediterranean model (MED) with a downscaling factor of 5 and the aim is to prove that the upscaling technique is driving the parent model (MED) solution towards the child model one (NW-Med). The upscaling consists in assimilating the 3D temperature and salinity child model fields as pseudo-obs in the parent model. The upscaled model solution is thus closer to the child model when compared to the parent model using 5 different metrics.

**General Comment**

The paper presents the upscaling technique as a relevant scientific question in the operational model community. After the revision the paper improved but there are things that reveal again a superficial approach of the corresponding author (I am sorry to say), which ignored some suggestions. Some corrections were only partially included. Added material (i.e. Tab.1 and the appendix) has not been described or motivated properly in the text.

I provide further suggestions to improve the paper readability and preciseness, with very detailed indications that required once again a lot of efforts.

However after these corrections the paper could be published.

**Specific Comments**

**Abstract**

**Lines 2-6-20, line 4-5-8-26 page 2,….**

I **already** suggested to revise the following nomenclature to be consistent with CMEMS one. In CMEMS (see Simoncelli et al., 2017!!!!) the **regional** models are considered the basin scale, while you use it to indicate the high resolution model. I seriously recommend to harmonize it in all the paper since you are in a CMEMS special issue.

Line 7→ please substitute "forecasts" with simulations, since you are not using the model in forecast mode. Again here you use low-resolution vs high resolution, other time you use parent and child, please harmonize.

Line 10: Being *in some sense* (not very scientific expression) *more realistic* means to have better prediction skills from model validation with observations, this should be specified.

Line 12: here you use "stand alone model" instead of introducing the **MED free model** and you could also specify the **MED upscaled model**. However you are stating that you are going to compare **MED free model** and **MED upscaled mode**l, but in the text and some figure you refer to NW MED model (i.e. Fig.9 and 10) and the NW upscaled MED model (i.e.Fig.11, Tab.1). This is another point to be harmonized, and that was ignored by the author.

Line 13: Looking at Tab. 1 you have improvements, if you consider improvement=decrease of RMSD on your metrics computed among parent-child models, that goes from 14% cross-shelf transp to 0% for SST. After SST Rhone plume metrics presents the smallest improvement. I expect/ed from the author this kind of evaluation which is totally missing in this second version. Tab. 1 has just been inserted without any explanation. I think that the paper should be full revised accordingly.

**Intro**

Line 19 page 1: I would take out "*regional and coastal*" referred to the oceanographic centers

Line 21 page 1: I would take out *"increased experience"*
Line 21 page 1**:** now you use *"local"* models …

Line 1 page 2: I suggest "*high resolution observations of currents*"

line 4 page 2**:** Again please use either low vs high resolution or parent vs child

line 7 page 2: Again….please use either low vs high resolution or parent vs child and substitute forecast with simulation.

line 9 page 2: *"This constitutes the baseline hypothesis of the present study: it is desirable to "copy" the results of the nested model into the parent model."*
Your **assumption** is that the child model performs better than the parent model within the child domain, your **objective** it to "copy/transfer/mimic?" the child model results in the parent one.

Line 11 and 17 page 2 → I would substitute *"forecasts"* with "*data"*

Line 12 page 3: I suggest to erase *"(only the horizontal grid is different). This could influence the conclusion compared to a set-up with 2 different model codes. However, this is not expected to be a fundamental limitation of the method. Concerning the vertical grid, in the usual case of assimilating real observations such as vertical profiles, the observations and 15 model forecasts have different vertical resolutions. Similarly, if the child model were on a different vertical grid than the parent model, it would still contain useful information, worth to be assimilated in the parent model. A limitation could be that some of the observations may be lost, e.g. the lowest child model layer may be out-of-grid in the parent model."*
The text is confusing but the content is obvious and does not justify 7 lines of text.

Line 12 page 3: I would substitute *"forecasts"* with "*data"*

Line 18 page 3: "If different model codes were used, the mode**l**s could represent different processes."

The reviewer considers the text lines 12-29 superficial. Independently from the models' set up, you use child model data as synthetic observations in your data assimilation scheme. As for any other type of observation the assimilation approach is tuned accordingly. Most important would be the model data thinning or weighting as function of child model skill, but the author highly underestimated this aspect, preferring a pure assimilation exercise approach.

Line 1 page 4: I would substitute *"forecasts"* with "*data"*

Line 1 page 4: *"It should be noted that some high-resolution processes, resolved by the nested model but not by the parent model, could have large phase errors in the nested model. In this case, the baseline hypothesis would be violated, and the nested model could actually have higher errors than the former. This aspect is not considered in the paper."*
Your SST metrics prove that you are in  this case thus I would avoid the last phrase and I would ameliorate your results description and Conclusions accordingly.

**2.1 Hydrodynamic Model**
**Line 12** page 4: Please refer to figure 2. I would also use " The region is characterized…"
Line 14 page 4: introduce the acronym at line 11 please.
Line 20 page 4: Please specify the resolution of the child model and correct the parent model resolution. 6 or 8 km? (See table in the appendix).

I do not agree on the choice of having an appendix with a table of identical columns. If you want to keep the table just mention it as Tab.1 here. Child and Parent model differences are the horizontal resolution 8km (not 6?), a highest resolution topography and bathymetry of the child model and the Rhone river discharge data.

Line 16 page 4: Please correct the reference in the bibliography to properly cite a specific section in the CMEMS OSR.
*Simoncelli, S., Pinardi, N., Claudia Fratianni, Dubois, C., Notarstefano, G. 2018. Water mass formation processes in the Mediterranean Sea over the past30 years. In: Copernicus Marine Service Ocean State Report, Issue 2, Journal of Operational Oceanography, 11:sup1, s13–s16, DOI: 10.1080/1755876X.2018.1489208*

Line 21 page 4
Please insert the references as suggested previously.

Med analyses at 1/16th
*Clementi E., Pistoia J., Fratianni C., Delrosso D., Grandi A., Drudi M., Coppini G., Lecci R., Pinardi N. (2017). Mediterranean Sea Analysis and Forecast (CMEMS MED-Currents 2013-2017). [Data set]. doi: https://doi.org/10.25423/MEDSEA_ANALYSIS_FORECAST_PHYS_006_001.*

Paper describing the reanalysis set up
*Simoncelli S., Masina S., Axell L., Liu Y., Salon S., Cossarini G., Bertino L., Xie J., Samuelsen A., Levier B., et al. (2017). MyOcean regional reanalyses: overview of reanalyses systems and main results. Mercator Ocean J. 54. Special Issue on Main Outcomes of the MyOcean2 and MyOcean Follow-on projects. [https://www.mercator-ocean.fr/wp-content/uploads/2017/04/Mercator-Ocean-newsletter-2015_54.pdf](https://www.mercator-ocean.fr/wp-content/uploads/2017/04/Mercator-Ocean-newsletter-2015_54.pdf)*

*Reanalysis data set*
*Simoncelli S, Fratianni C, Pinardi N, Grandi A, Drudi M, Oddo P, Dobricic S. 2014. Mediterranean Sea physical reanalysis (MEDREA 1987-2015) [dataset]. Copernicus Monitoring Environment Marine Service (CMEMS). doi:10.25423/medsea_reanalysis_phys_006_004.*

Line 24-27 page 4: this is redundant, it's already written in the introduction.

Line 30 page 4: How do you interpolate MED reanalysis data onto your parent and child model grid? Or is it only for the child model? Which kind of extrapolation did you apply where model topographies mismatch? i.e. Coastal strip, or bottom layers deeper than MED reanalysis ones.

Line 11 page 5: I suggest "…showing the surface salinity difference using climatological or daily data in child model (NW MED model) simulations after 1 month of spin up"

**2.2**
Line 24 Page 5: none detail is in the annex about the data assimilation. Modify accordingly.
Line 25: whole or thinned?
Line 26 isn't is a super-obs approach?

You are mixing the description of data assimilation and initial condition, I recommend to start from the upscaling experiment description (absent now), then IC and then DA. Moreover in Tab 1 you refer to 2 nested systems, thus you should explain both experiments.

Are the perturbed IC applied to both child and parent models or only the child, this is not specified.

**3. Metrics**

Please introduce Tab 1 and its interpretation either here or in **4. Results**. Now you mention it in the last line of section 4.5. Insert its reference also in all metrics discussions in 4.*.

**3.4 "***This metric is the root mean square (rms) difference between the model and observed SST. For the latter, the L3 images are used. *"**
This paragraph could be improved, among which models? What is in Tab.1? How is it computed? I suggest also to remove the second phrase, you are not talking about this afterwards.

**3.5** I thank the author for the explanation however the text has not been modified. I suggest to do so, without mentioning the tail of the diagram. The reader would thank you.

**4. Results**
Figure 4: please increase the size of the red arrow

Figure 5: avoid to use forecast (plot titles), use consistent nomenclature in the caption.

**4.2** second line, I would use *child* instead of *nested (same in caption of Figure 3*) as in the rest of the paper to harmonize and facilitate the reader. (Already suggested)

**4.3**
First Line: Why do you say that? Why don't you use Tab.1 to argument your statement?
Lines 3-4: The interpretation of Fig.8 is confusing. The upper panel shows the **MED free model**, please change the title in agreement with figures 6, 7. The bottom panel shows the **MED upscaled model**, please change the title in the plot accordingly. Why don't you comment the **MED upscaled model** instead of the nested model? You do it at line 2 of page 13
Moreover the arrows are pointing North-West or South West, is it correct? Could you better explain and interpret the figure for the reader?
I suggest to revise the paragraph and adopt the same nomenclature in figures/captions/text. This suggestion was not handled by the author.

Figure 9: This figure presents MED and NW-Med, why not the **MED upscaled model**? I suggest to show the three salinity fields. The author just skipped this suggestion, however the reader is confused since you always change approach in presenting the results. The nomenclature in Fig. 9 is not consistent, please change the titles to match **MED free model** and **MED upscaled model.** The author replied that the **MED upscaled model** is indistinguishable from the nested model but the scope here should be to show that the **MED upscaled model** is close to the nested/child model and not that the nested model is closer to the satellite image. From my point of view the author's answer is very superficial.

Line 10-13 Page 13: Considering that you do not care about what observations indicate, you say that upscaling is changing in-depth salinity in the ECC and WCC. This phrase should start a new line because not related to the Rhone plume, otherwise please explain what is the connection and motivate why upscaling is behaving in the right direction.

**4.4**
Line 3 Page 14: *Level 3 images are used for computing the metric*→ already said in 3.4
*(a level 4 image shown in Fig. 5 is used only for visual comparison)* → This should not go here but in the Fig. 5 caption and specified at line 26 page 9.

Line 4 Page 14: "Results are shown in Fig. 10." What is the plot? What do you want to show? You say it at line 4 Page 15: "*Fig. 10 shows the RMS error during the first 2 months of simulation.*" → of what, which models????

My suggestion is to revise the entire paragraph.

Line 4 Page 14: I do not agree that the **MED free model** is in very good agreement with SST, at least you do motivate it, including some reference to support it. What is the CMEMS skill in this region/period? http://cmems-resources.cls.fr/documents/QUID/CMEMS-MED-QUID-006-013.pdf
In fact in the paragraph You say that the error is relatively large in some days, that during summer is around 3 degrees C, that all the models are not resolving some coastal processes.

Line 5 Page 14: *"Usually, the nested model is better still in some areas (e.g. coastal waters), and the upscaling procedure brings back these local improvements to the parent model."* Please rephrase, this statement is vague. You assume that the nested model is performing better in coastal waters, thus your technique should modify the parent model and increase its performance accordingly, right?

Line 5 Page 15:  *The situation worsens during summer when the computed RMS errors are of 3_C, both for parent and child model* (not shown).
It goes at the beginning of the paragraph.

*"The difference in between models is hidden by the temporal variability of the error. In any case, the upscaled model is still very close, and slightly better, than both the (free) parent and the nested models."*
From my point of view, there are not differences among the **MED free model, MED upscaled model and NW-MED model (Tab.1 prove it).** Please provide the average RMS computed over the considered time period, if you want to say that **MED upscaled model is slightly better than MED free and NW-MED.**

Line 8-13: They are about the model temperature in depth and should not go in this paragraph, eventually in the general discussion of results or in the summary.
*"The model temperature in depth can be only punctually evaluated against observations (when e.g. drifter observations are available). In any case, the goal of the current study is to check whether upscaling is able to bring the parent model closer to the nested model, under the hypothesis that the latter is "better" in some sense.* (not needed here it's a repetition). *Differences between the parent and the nested model are locally important, e.g. on the bottom of the Gulf of Lions, or in the Eastern Corsican and Northern Current cores (with differences of up to 0.3_C), and upscaling is able to push the temperature field in the parent model toward the nested model solution."* (not pertinent here and not shown!).

**4.5**
Again, what are you showing in Figure 10? I suggest also to modify:
*"The total amount of Western Mediterranean Deep Water in the free model (blue curve in Fig. 11) and the nested model (green curve) is periodically important (103 km3 ),  but the-models do not  converge during the simulation.  , as it appears during most of the second half of the year."*

Line 4 Page 16: I would use the reference to Tab. 1 here instead of line 9.

**5. Conclusions**
Lines 10 Page 17: You should say that for SST the upscaling did not produce any improvement, as shown also by Tab.1

In fact, you say in 4.4.

*"The  spatial RMS error is not influenced  by upscaling (please refer to Tab.1), as large areas are essentially unmodified (parent and child models use the same atmospheric forcing fields). Some days, some processes  are not resolved by the models (both parent and nested), so that the RMS error is relatively large. In this case again (?), upscaling does not influence the RMS error of the parent model  (the RMS in tab1 is identical), as the nested model is not representing these processes any better than the parent model."*

This suggest that without considering the skill of the child model, your upscaling might not improve the parent model solution, but just bring the child solution closer to the parent one blindly. It could also degrade the parent model performance. Obviously if you do not validate the models with observations, you do not know.

---

## Referee Report (RR2)

**2nd Review**
Research article os-2018-79 entitled:
Upscaling of regional models into basin-wide models,
by Vandenbulcke Luc and Barth Alexander.

Recommendation - accept.

The authors thoroughly addressed all my comments and changed the manuscript accordingly. Therefore, I recommend accepting the revised manuscript for publication in Ocean Science.

Sincerely yours.

---

## Author Response (AR2)

Dear Madam,

We thank the first anonymous reviewer for recommanding to accept the article as is, and the second reviewer for yet again precise and helpful comments. Clearly, she/he spent again a lot of time trying to help us get a very good manuscript.

We answered each of his/her comments, and modified the text and the figures accordingly. The revised version of the manuscript is much more consistent, some new references were again added, all according to the reviewer's comments.

The second reviewer also suggested modifying the « regional » to « nested », as in CMEMS terminology, regional means basin-wide, whereas by regional model, we understood a « local » model. We modified the paper accordingly, but also the title of the paper should now be changed, « regional » becoming « nested ».

We again think the revised paper is much better than the previous version, and submit it to you for consideration for publication in OS.

The replies to the second reviewer are copied below for your convience.

**Upscaling of regional models into basin-wide models**

**Luc Vandenbulcke and Alexander Barth**

Research article in Special Issue: The Copernicus Marine Environment Monitoring Service (CMEMS): scientific advances

The paper presents the upscaling technique in a realistic configuration in the Mediterranean Sea domain. A NW-Med model is nested into a Mediterranean model (MED) with a downscaling factor of 5 and the aim is to prove that the upscaling technique is driving the parent model (MED) solution towards the child model one (NW-Med). The upscaling consists in assimilating the 3D temperature and salinity child model fields as pseudo-obs in the parent model. The upscaled model solution is thus closer to the child model when compared to the parent model using 5 different metrics.

**General Comment**

The paper presents the upscaling technique as a relevant scientific question in the operational model community. After the revision the paper improved but there are things that reveal again a superficial approach of the corresponding author (I am sorry to say), which ignored some suggestions. Some corrections were only partially included. Added material (i.e. Tab.1 and the appendix) has not been described or motivated properly in the text.

I provide further suggestions to improve the paper readability and preciseness, with very detailed indications that required once again a lot of efforts.

However after these corrections the paper could be published.

We thank the reviewer for the constructive comments and the obviously large amount of work that went in the review. We answered each comment in the document below and corrected the article accordingly.

**Specific Comments**

**Abstract**

**Lines 2-6-20, line 4-5-8-26 page 2,....**

I already suggested to revise the following nomenclature to be consistent with CMEMS one. In CMEMS (see Simoncelli et al., 2017!!!!) the **regional** models are considered the basin scale, while you use it to indicate the high resolution model. I seriously recommend to harmonize it in all the paper since you are in a CMEMS special issue.

About the use of the word « regional »... We do not agree with the reviewer that « regional model » should necessarily mean a basin-scale model. Lorente et al, JOO 2016, talks about the IBI model in CMEMS and uses « region » to talk about a specific sub-domain of IBI. In general, a region may refer to both large (e.g. 'Eastern Europe') but also to smaller entities (e.g. the 20 regions of Italy). However, we want to avoid any potential misunderstanding, and it's true that the CMEMS website talks about « regions » for the 7 european seas. To be coherent with CMEMS' choice for the word « region », we do not refer to the NW-Med model as « regional » anymore. We now talk about the « nested model », « child model », « high-resolution model » instead of « regional model », and introduce the « sub-regional » term instead of « regional » when talking about the geographical area. To be coherent, we also need to modify the title of the paper, which became « upscaling of nested models... »

Line  $7 \rightarrow$  please substitute "forecasts" with simulations, since you are not using the model in forecast mode.

We replaced « forecast » with simulation

**Again here you use low-resolution vs high resolution, other time you use parent and child, please harmonize.**

It is true that we use both « parent model » and « low-resolution model » when talking about the Med model ; and « child model », « nested model » and « high-resolution model » when talking about the NW-Med model. In our opinion, it is pretty obvious which model is referred to, but to be even more clear, we now explicitly explain this in the introduction. Using only « high-resolution model » (and repeating it over and over again) in the paper would lead to an extremely boring text for a paper that is already kind of « technical ».

Line 10: Being in some sense (not very scientific expression) more realistic means to have better prediction skills from model validation with observations, this should be specified. Changed «... is, in some sense, more realistic ... » to « ... has better prediction skills ... »

Line 12: here you use "stand alone model" instead of introducing the **MED free model** and you could also specify the **MED upscaled model**. However you are stating that you are going to compare **MED**

**free model** and **MED upscaled model**, but in the text and some figure you refer to NW MED model (i.e. Fig.9 and 10) and the NW upscaled MED model (i.e. Fig.11, Tab.1). This is another point to be harmonized, and that was ignored by the author.

We removed « standalone model » as it was the only place in the article that this was used, and added a note at the bottom of the introduction instead. Also «free model » and « upscaled model » are now introduced in the bottom of the introduction.

Also, in general, we tried to remove the remaining lack of harmony in the figures and table.

Line 13: Looking at Tab. 1 you have improvements, if you consider improvement=decrease of RMSD on your metrics computed among parent-child models, that goes from 14% cross-shelf transp to 0% for SST. After SST Rhone plume metrics presents the smallest improvement. I expect/ed from the author this kind of evaluation which is totally missing in this second version. Tab. 1 has just been inserted without any explanation. I think that the paper should be full revised accordingly. We added some text in the relevant sub-sections of section 4 (Results) explaining better the results summarized in Table 1. Also in the conclusion a sentence was added.

**Intro**

Line 19 page 1: I would take out "regional and coastal" referred to the oceanographic centers Removed

Line 21 page 1: I would take out "increased experience" OK, removed

Line 21 page 1: now you use "local" models ... Changed to « nested models »

Line 1 page 2: I suggest "high resolution observations of currents" Changed

line 4 page 2: Again please use either low vs high resolution or parent vs child changed to « parent and child »

line 7 page 2: Again....please use either low vs high resolution or parent vs child and substitute forecast with simulation.

Changed to « parent and child », and replaced forecast by « model »

line 9 page 2: "This constitutes the baseline hypothesis of the present study: it is desirable to "copy" the results of the nested model into the parent model."

Your assumption is that the child model performs better than the parent model within the child domain, your objective it to "copy/transfer/mimic?" the child model results in the parent one. Yes

Line 11 and 17 page 2 à I would substitute "forecasts" with "data"

Line 11 : we changed forecasts to « results »

Line 17 : we changed forecasts to « data »

As a side-note, the « forecasts » of the nested model COULD be extracted, and used as « future pseudo-observations » in a re-run of the parent model. This is the only case where we actually have « future observations » to assimilate in forecast mode.

But this is not considered in the present paper. In the paper, we don't pay attention to the fact that we run in hindcast or forecast. So we agree with the reviewer and remove the word « forecast ».

Line 12 page 3: I suggest to erase "(only the horizontal grid is different). This could influence the conclusion compared to a set-up with 2 different model codes. However, this is not expected to be a fundamental limitation of the method. Concerning the vertical grid, in the usual case of assimilating real observations such as vertical profiles, the observations and 15 model forecasts have different vertical resolutions. Similarly, if the child model were on a different vertical grid than the parent model, it would still contain useful information, worth to be assimilated in the parent model. A limitation could be that some of the observations may be lost, e.g. the lowest child model layer may be out-of-grid in the parent model."

The text is confusing but the content is obvious and does not justify 7 lines of text.

This text was added following a question of the other reviewer, « how would the method behave if the nested model was not NEMO or had a different vertical grid ». It is now replaced with a single sentence : *It is not expected that the conclusions of the study would be fundamentally different if different models and vertical grids are used for parent and child models.*

Line 12 page 3: I would substitute "forecasts" with "data" Removed, there is not any single « forecast » in the article anymore

Line 18 page 3: "If different model codes were used, the models could represent different processes."

The reviewer considers the text lines 12-29 superficial. Independently from the models' set up, you use child model data as synthetic observations in your data assimilation scheme. As for any other type of observation the assimilation approach is tuned accordingly. Most important would be the model data thinning or weighting as function of child model skill, but the author highly underestimated this aspect, preferring a pure assimilation exercise approach.

Part of the text is removed (see previous comment). The remainder answers a question from the other reviewer, and is (also in our opinion) useful.

Regarding the reviewer's comment on the data thinning and weighting as a function of the nested model skill, this implies to validate the child model with real observations. If real observations are available in sufficient quantity, they would probably be sufficient to constrain the parent model as well, i.e. they could be assimilated directly in the parent model. The exception to this, is the case where very dense (in space) observations are available, that could better be ingested by the nested model than by the parent model (e.g. ultra-high-res SST, or radar surface currents).

Line 1 page 4: I would substitute "forecasts" with "data" Modified « forecasts » into « simulation »

Line 1 page 4: "It should be noted that some high-resolution processes, resolved by the nested model but not by the parent model, could have large phase errors in the nested model. In this case, the baseline hypothesis would be violated, and the nested model could actually have higher errors than the former. This aspect is not considered in the paper."

Your SST metrics prove that you are in this case thus I would avoid the last phrase and I would ameliorate your results description and Conclusions accordingly.

We do not agree with the reviewer that the SST metric proves that we are in the case of phase errors in the nested model. Typically, phase errors are visible e.g. in inertial oscillations, after a wind burst. But in the current experiment, for the SST field, if the parent and child models both have small errors (say, smaller than 1°) and at another period, both have large errors (say, 3°C), then I would rather

suspect errors in the atmospheric forcing fields and in the bulk formulae computing the heat fluxes, which are the same in both models. Also the heat propagation towards depth uses the same attenuation parameters in both models.

**2.1 Hydrodynamic Model**

Line 12 page 4: Please refer to figure 2. I would also use "The region is characterized..." Added the reference, and changed the text according to the comment

Line 14 page 4: introduce the acronym at line 11 please. Added

Line 20 page 4: Please specify the resolution of the child model and correct the parent model resolution. 6 or 8 km? (See table in the appendix).

8 km, sorry for the confusion, corrected now

I do not agree on the choice of having an appendix with a table of identical columns. If you want to keep the table just mention it as Tab.1 here. Child and Parent model differences are the horizontal resolution 8km (not 6?), a highest resolution topography and bathymetry of the child model and the Rhone river discharge data.

The reviewer is right, and the appendix with the table has now been removed. The important differences between child and parent models are given around line 20 page 4.

Line 16 page 4: Please correct the reference in the bibliography to properly cite a specific section in the CMEMS OSR. Simoncelli, S., Pinardi, N., Claudia Fratianni, Dubois, C., Notarstefano, G. 2018. Water mass formation processes in theMediterranean Sea over the past30 years. In: Copernicus Marine Service Ocean State Report, Issue 2, Journal of Operational Oceanography, 11:sup1, s13–s16, DOI: 10.1080/1755876X.2018.1489208 corrected

Line 21 page 4 Please insert the references as suggested previously. Med analyses at 1/16th Clementi E., Pistoia J., Fratianni C., Delrosso D., Grandi A., Drudi M., Coppini G., Lecci R., Pinardi N. (2017). Mediterranean Sea Analysis and Forecast (CMEMS MED-Currents 2013-2017). [Data set]. doi:https://doi.org/10.25423/MEDSEA\_ANALYSIS\_FORECAST\_PHYS\_006\_001. Added

Paper describing the reanalysis set up Simoncelli S., Masina S., Axell L., Liu Y., Salon S., Cossarini G., Bertino L., Xie J., Samuelsen A., Levier B., et al. (2017). MyOcean regional reanalyses: overview of reanalyses systems and main results. Mercator Ocean J. 54. Special Issue on Main Outcomes of the MyOcean2 and MyOcean Follow-on projects. https://www.mercator-ocean.fr/wp-content/uploads/2017/04/Mercator-Ocean-newsletter-2015\_54.pdf

Reanalysis data set

Simoncelli S, Fratianni C, Pinardi N, Grandi A, Drudi M, Oddo P, Dobricic S. 2014. Mediterranean Sea physical reanalysis (MEDREA 1987-2015) [dataset]. Copernicus Monitoring Environment Marine Service (CMEMS). doi:10.25423/medsea\_reanalysis\_phys\_006\_004.

Added

Line 24-27 page 4: this is redundant, it's already written in the introduction. The general idea has been already written, but the particular impact of upscaling on the Corsican currents and the stratification has not been mentionned before

Line 30 page 4: How do you interpolate MED reanalysis data onto your parent and child model grid? Or is it only for the child model? Which kind of extrapolation did you apply where model topographies mismatch? i.e. Coastal strip, or bottom layers deeper than MED reanalysis ones. Tri-linear interpolation and linear extrapolation, this is now written in the paper

Line 11 page 5: I suggest "...showing the surface salinity difference using climatological or daily data in child model (NW MED model) simulations after 1 month of spin up" Changed the text as suggested by the reviewer

**2.2**

Line 24 Page 5: none detail is in the annex about the data assimilation. Modify accordingly. Modified Line 25: whole or thinned?

What I meant is : the whole 3D field, but thinned. Rewritten as : the thinned 3D field Line 26 isn't is a super-obs approach? I guess so...

You are mixing the description of data assimilation and initial condition, I recommend to start from the upscaling experiment description (absent now), then IC and then DA. The title of the subsection indicates what will be described : «Upscaling experiment description, Ensemble generation, and Data Assimilation scheme ». This is what the reviewer also suggests. The 3 paragraphs in the subsection are now better separated to reflect that. The first paragraph is a description of the upscaling experiment.

Moreover in Tab 1 you refer to 2 nested systems, thus you should explain both experiments. This is explained at the very beginning of section 3. This text has been slightly re-written to be more clear.

Are the perturbed IC applied to both child and parent models or only the child, this is not specified. The perturbed IC (as well as the other perturbations) are applied only to the parent model ; upscaling consists in assimilating into the *parent* model. As a EnKF is used, the parent model thus needs to be transformed into an ensemble. This is now briefly reminded in the description of the upscaling experiment (i.e. at the top of this subsection)

**3. Metrics**

Please introduce Tab 1 and its interpretation either here or in **4. Results**. Now you mention it in the last line of section 4.5. Insert its reference also in all metrics discussions in 4.\*. We now refer to the Table in each subsection of section 4 (results)

**3.4** "This metric is the root mean square (rms) difference between the model and observed SST. For the latter, the L3 images are used. <del>Furthermore, in order to examine the position of features such as fronts and eddies, the rms difference of the norm of the spatial gradient of the SST is also</del>

**computed."**

This paragraph could be improved, among which models? What is in Tab.1? How is it computed? I suggest also to remove the second phrase, you are not talking about this afterwards. Indeed. The second phrase is now removed.

The RMS is computed between the parent model and the L3 SST image (for both the free and upscaled parent model). This is now written in the text.

3.5 I thank the author for the explanation however the text has not been modified. I suggest to do so, without mentioning the tail of the diagram. The reader would thank you. The text has now been adapted, and the whole issue of the tail of the diagram has been deleted from the text.

**4. Results**

Figure 4: please increase the size of the red arrow Done

Figure 5: avoid to use forecast (plot titles), use consistent nomenclature in the caption. Modified plot titles and changed nomenclature to « MED free », « NW-Med » and « MED upscaled » identical to the other plots

**4.2** second line, I would use child instead of nested (same in caption of Figure 3) as in the rest of the paper to harmonize and facilitate the reader. (Already suggested) Changed

**4.3**

First Line: Why do you say that? Why don't you use Tab.1 to argument your statement? Table 1 shows the RMS for the plume length. The plume direction (offshore or along-shore) is sometimes different in the parent and child model. When upscaling modifies that (succesfully), it is a very significant change, although the RMS does not necessarily reflect that. The text was rephrased to better explain that.

Lines 3-4: The interpretation of Fig.8 is confusing. The upper panel shows the MED free model, please change the title in agreement with figures 6, 7. The bottom panel shows the MED upscaled model, please change the title in the plot accordingly. Why don't you comment the MED upscaled model instead of the nested model? You do it at line 2 of page 13

The figure titles were changed

The text is changed. In the paper, it now describes the « MED free » plume, then the « NW-MED and MED upscaled» plume.

Moreover the arrows are pointing North-West or South West, is it correct? Could you better explain and interpret the figure for the reader?

The reviewer is absolutely right, I actually meant to say « West » and wrote « East » for both cases. This is now corrected in the text.

I suggest to revise the paragraph and adopt the same nomenclature in figures/captions/text. This suggestion was not handled by the author.

The nomenclature is now changed and consistent with the rest of the paper and the other figures.

Figure 9: This figure presents MED and NW-Med, why not the MED upscaled model? I suggest to

show the three salinity fields. The author just skipped this suggestion, however the reader is confused since you always change approach in presenting the results.

We replaced now the NW-MED plot with the MED upscaled plot (projected on the NW-MED grid), as the reviewer suggests. It's true this is more coherent.

The nomenclature in Fig. 9 is not consistent, please change the titles to match MED free model and MED upscaled model.

Changed according to the reviewer's suggestion

The author replied that the MED upscaled model is indistinguishable from the nested model but the scope here should be to show that the MED upscaled model is close to the nested/child model and not that the nested model is closer to the satellite image. From my point of view the author's answer is very superficial.

The reviewer is right, and we hope the new plots are more convincing. The chlorophyll plot is still left in the paper, as illustration.

Please note that both plots were from Nemo restart files (in the previous version of the submitted paper), and now they are daily means. I don't have anymore the corresponding restart file for the MED upscaled. This does not change anything, but I mention it to explain why very small differences appear also in the MED free plot (compared to the previous version of the paper).

Line 10-13 Page 13: Considering that you do not care about what observations indicate, you say that upscaling is changing in-depth salinity in the ECC and WCC. This phrase should start a new line because not related to the Rhone plume, otherwise please explain what is the connection and motivate why upscaling is behaving in the right direction.

This is indeed unrelated to the Rhone plume. It was moved into a separate paragraph

**4.4**

Line 3 Page 14: Level 3 images are used for computing the metric  $\rightarrow$  already said in 3.4 (a level 4 image shown in Fig. 5 is used only for visual comparison)  $\rightarrow$  This should not go here but in the Fig. 5 caption and specified at line 26 page 9. Moved

Line 4 Page 14: "Results are shown in Fig. 10." What is the plot? What do you want to show? You say it at line 4 Page 15: "Fig. 10 shows the RMS error during the first 2 months of simulation." à of what, which models????

My suggestion is to revise the entire paragraph.

The entire paragraph has been rewritten and figures are clearly described

Line 4 Page 14: I do not agree that the **MED free model** is in very good agreement with SST, at least you do motivate it, including some reference to support it. What is the CMEMS skill in this region/period? http://cmems-resources.cls.fr/documents/QUID/CMEMS-MED-QUID-006-013.pdf In fact in the paragraph You say that the error is relatively large in some days, that during summer is around 3 degrees C, that all the models are not resolving some coastal processes. During the first 2 months, the RMS between model and SST is around 0.4 or 0.5°C **without data assimilation**, and in my opinion, this is not bad. During the remainder of the simulation, and especially during summer, the model is less good, and indeed errors go up to 3°C, which is not good. Rather than make a quality document of the model, which is not our aim, the « very good agreement » has been removed.

Line 5 Page 14: "Usually, the nested model is better still in some areas (e.g. coastal waters), and the upscaling procedure brings back these local improvements to the parent model." Please rephrase, this statement is vague. You assume that the nested model is performing better in coastal waters, thus your technique should modify the parent model and increase its performance accordingly, right? The method does not assume geographical criteria (coastal or not, etc). But it so happens that the nested model sometimes performs better at the shelf break or on the shelf. These improvements are indeed brought back to the « MED upscaled » model. When the rest of the domain is essentially unmodified, this improvement at the shelf break almost doesn't change the overall RMS. We added a reference to Figure 5, which shows an example for a coastal or shelf-break process improved by upscaling.

Line 5 Page 15: A similar plot for the whole of 2014 shows that The situation worsens during summer when the computed RMS errors are of 3\_C, both for parent and child model (not shown). It goes at the beginning of the paragraph.

We changed the text according to the reviewer's suggestion, but prefer to keep the description of the 2 first months (and the corresponding figure) separated from the remainder of the year.

"The difference in between models is hidden by the temporal variability of the error. In any case, the upscaled model is still very close, and slightly better, than both the (free) parent and the nested models."

From my point of view, there are not differences among the **MED free model**, **MED upscaled model** and **NW-MED model (Tab.1 prove it)**. Please provide the average RMS computed over the considered time period, if you want to say that **MED upscaled model is slightly better than MED** free and NW-MED.

We removed the « slightly better », even though the 1.3°C RMS error is actually very slightly smaller for the upscaled model than the free model (when not rounding the RMS).

Line 8-13: They are about the model temperature in depth and should not go in this paragraph, eventually in the general discussion of results or in the summary.

"The model temperature in depth can be only punctually evaluated against observations (when e.g. drifter observations are available). In any case, the goal of the current study is to check whether upscaling is able to bring the parent model closer to the nested model, under the hypothesis that the latter is "better" in some sense. (not needed here it's a repetition). Differences between the parent and the nested model are locally important, e.g. on the bottom of the Gulf of Lions, or in the Eastern Corsican and Northern Current cores (with differences of up to 0.3\_C), and upscaling is able to push the temperature field in the parent model toward the nested model solution." (not pertinent here and not shown!).

The reviewer is right, and actually the same thing happens in the previous section, which uses surface salinity to describe the Rhone plume, but then quickly also says a word about in-depth salinity. Both for salinity and temperature, the paragraphs have been moved into a new, separated but un-numbered subsection (after the 5 subsections corresponding to the 5 metrics).

Furthermore, the text was modified according to the reviewer's suggestions, and generally simplified.

\subsection\*{Deep temperature and salinity}

Some metrics considered above used surface salinity and temperature. However, upscaling modifies the 3D variables of temperature and salinity.

Differences between the parent and the nested model temperature are locally important, e.g. on the bottom of the Gulf of Lions, or in the Eastern Corsican and Northern Current cores (with differences of

up to 0.3°C). Similarly, the cores of both Corsican Currents are saltier in the upscaled model, with differences of about 0.15 psu during the first assimilation cycle. For both temperature and salinity, upscaling is able to push the parent model toward the child model solution (not shown).

**4.5**

Again, what are you showing in Figure 10?

For fig.10 this is now described in the text. Fig.11 is already described in the text.

I suggest also to modify:

"The total amount of Western Mediterranean Deep Water in the free model (blue curve in Fig. 11) and the nested model (green curve) is periodically important (103 km3), <del>and both</del> but the models do not <del>appear</del> to converge during the simulation. <del>On the contrary, a period of large discrepency</del>, as it appears during most of the second half of the year."

The text has been modified according to the suggestion

Line 4 Page 16: I would use the reference to Tab. 1 here instead of line 9. Moved the reference to Tab. 1

**5. Conclusions**

Lines 10 Page 17: You should say that for SST the upscaling did not produce any improvement, as shown also by Tab.1

This paragraph of the conclusion was rewritten, it is now mentioned that the SST RMS is not improved. It is also mentioned that some local improvements (such as seen in Fig. 5) are not contributing significantly to the RMS (as explained already in 4.4).

In fact, you say in 4.4.

"The area-wide spatial RMS error is not influenced very much by upscaling (please refer to Tab.1), as large areas are essentially unmodified (parent and child models use the same atmospheric forcing fields). Some days, some processes appear to be missed are not resolved by the models (both parent and nested), so that the RMS error is relatively large. In this case again (?), upscaling does not influence the RMS error of the parent model very much (the RMS in tab1 is identical), as the nested model is not representing these processes any better than the parent model."

The « area wide » is important to underline that the RMS is computed over the domain, whereas the improvements (when there are any) are only local (e.g. coastal etc, see before). Therefor, the RMS is not improved. This does not imply that there are no changes and that some processes are not modified by upscaling.

Anyway, the reference to Tab. 1 is now added in the sentence as suggested by reviewer 1.

This suggest that without considering the skill of the child model, your upscaling might not improve the parent model solution, but just bring the child solution closer to the parent one blindly. Yes. Or more exactly, we bring the parent solution closer to the child. This was exactly our aim. After that, it is up to modellers to decide if that's what they actually want. The decision could be based on observations, as the reviewer keeps suggesting, or on other knowledge.

It could also degrade the parent model performance. Obviously if you do not validate the models with observations, you do not know.

Indeed. As mentioned before, when one has lots of observations to validate / improve / assimilate in the parent model, the whole upscaling exercice becomes kind of pointless. But if there are (some) observations, of course it makes sense to validate the child model before considering it as pseudo-observations.

**Upscaling of nested models into basin-wide models**

Vandenbulcke Luc1,2 and Barth Alexander3

1seamod.ro, Jailoo srl, Romania
 2MAST, Université de Liège, Belgium

[revised manuscript text omitted]